# Separation of variables bases for integrable $gl_{\mathcal{M}|\mathcal{N}}$ and Hubbard models

**Jean Michel Maillet[⋆], Giuliano Niccoli[†] and Louis Vignoli[‡]**

Univ Lyon, ENS de Lyon, Univ Claude Bernard, CNRS,
Laboratoire de Physique, UMR 5672, F-69342 Lyon, France

⋆ maillet@ens-lyon.fr, † giuliano.niccoli@ens-lyon.fr, ‡ louis.vignoli@ens-lyon.fr

## Abstract

We construct quantum Separation of Variables (SoV) bases for both the fundamental inhomogeneous $gl_{\mathcal{M}|\mathcal{N}}$ supersymmetric integrable models and for the inhomogeneous Hubbard model both defined with quasi-periodic twisted boundary conditions given by twist matrices having simple spectrum. The SoV bases are obtained by using the integrable structure of these quantum models, i.e. the associated commuting transfer matrices, following the general scheme introduced in [1]; namely, they are given by set of states generated by the multiple actions of the transfer matrices on a generic co-vector. The existence of such SoV bases implies that the corresponding transfer matrices have non-degenerate spectrum and that they are diagonalizable with simple spectrum if the twist matrices defining the quasi-periodic boundary conditions have that property. Moreover, in these SoV bases the resolution of the transfer matrix eigenvalue problem leads to the resolution of the full spectral problem, i.e. both eigenvalues and eigenvectors. Indeed, to any eigenvalue is associated the unique (up to a trivial overall normalization) eigenvector whose wave-function in the SoV bases is factorized into products of the corresponding transfer matrix eigenvalue computed on the spectrum of the separated variables. As an application, we characterize completely the transfer matrix spectrum in our SoV framework for the fundamental $gl_{1|2}$ supersymmetric integrable model associated to a special class of twist matrices. From these results we also prove the completeness of the Bethe Ansatz for that case. The complete solution of the spectral problem for fundamental inhomogeneous $gl_{\mathcal{M}|\mathcal{N}}$ supersymmetric integrable models and for the inhomogeneous Hubbard model under the general twisted boundary conditions will be addressed in a future publication.



# 1 Introduction

In this paper, we generalize the construction introduced in [1] to generate quantum separation of variables (SoV) bases for the class of integrable quantum lattice models associated to the $gl_{\mathcal{M}|\mathcal{N}}$ Yang-Baxter superalgebras [2–4] and to the Hubbard model [5–10] with quasi-periodic twisted boundary conditions given by twist matrices having simple spectrum. The quantum version of the separation of variables and its development in the integrable framework of the quantum inverse scattering method [11–19] originate in the pioneering works of Sklyanin [20–25]. Since then, the SoV method has been successfully applied to several quantum integrable models [26–60] .

Integrable quantum models define the natural background to look for exact non-perturbative results toward the complete solution of some 1+1 dimensional quantum field theories or some equivalent two-dimensional systems in statistical mechanics. They have found natural applications in the exact description of several important phenomena in condensed matter and have provided exact results to be compared with experiments. A prominent example is the quantum Heisenberg spin chain [61] introduced as a model to study phase transitions and critical points of magnetic systems. First exact results for the Hamiltonian's spectrum (eigenvalues and eigenvectors) have been obtained by Bethe for the spin 1/2 XXX chain, thanks to his famous coordinate ansatz [62]. Then it has been extended to the anisotropic spin 1/2 XXZ chain in [63,64], while Baxter [65,66] has obtained first exact results for the Hamiltonian's spectrum of the fully anisotropic spin 1/2 XYZ chain. In statistical mechanics, these quantum models correspond to the six-vertex and eight-vertex models. The ice-type (six-vertex) models [67], accounting for the residual entropy of water ice for crystal lattices with hydrogen bonds, has

first been described in the Bethe Ansatz framework in [68]. For the eight-vertex model [69,70] first exact results in the two-dimensional square lattice are due to Baxter [66,71,72]. The integrable structure[1] of these spin chains and statistical mechanics models has been revealed in the Baxter's papers [65,72,78]. There, the one-parameter family of eight-vertex transfer matrices have been shown to be commutative and the XYZ Hamiltonian to be proportional to its logarithmic derivative, when computed at a particular value of its spectral parameter. The subsequent development of a systematic description of quantum integrable models has been achieved through the development of the quantum inverse scattering framework [11–19]. In particular, the work of Faddeev, Sklyanin and Takhtajan [12] has set the basis for the classification of the Yang-Baxter algebra representations and the natural framework for the discovering of quantum groups [79–82]. The paper [12] has also introduced the Algebraic Bethe Ansatz (ABA), an algebraic version of the original coordinate Bethe Ansatz.

Exact results are also available for the quantum dynamics, *i.e.* form factors and correlations functions, of some integrable quantum models. This is for example the case for XXZ quantum spin 1/2 chain under special boundary conditions whose correlation functions admit multiple integral representations [83–95].

In this context, the form factor expansion has proven to be a very powerful tool: on the one hand, to compute dynamical structure factors [96,97], quantities directly accessible experimentally through neutron scattering [98]; on the other hand, to have access to the asymptotic behaviour of correlation functions of these XXZ chains in the thermodynamic limit and explicit contact with conformal field theory [90–92,99–105].

Integrable quantum models also led to non-perturbative results in the out-of-equilibrium physics context, ranging from the relaxation behaviour of some classical stochastic processes to quantum transport. The XXZ quantum spin chains, under general integrable boundary conditions, appear for example both in the description of the asymmetric simple exclusion processes [106–112] and the description of transport properties of quantum spin systems [113,114].

The new experiments allowing ultra-cold atoms to be trapped in optical lattices have produced concrete realizations of quantum integrable lattices, like the Heisenberg spin chains but also more sophisticated models like the Hubbard model [115,116]. They provide a further natural context for direct comparison of exact theoretical predictions with experiments.

The Hubbard model is of fundamental importance in physics. It is a celebrated quantum model in condensed matter theory, defining a first generalization beyond the band approach for modelling the solid state physics. It manages to describe interacting electrons in narrow energy bands and allows to account for important physical phenomena of different physical systems. Relevant examples are the high temperature superconductivity, band magnetism and the metal-insulator transitions. We refer to the book [10] for a more detailed description of its physical applications and of the known exact results and relevant literature. Here, let us recall that the Hubbard chain is integrable in the quantum inverse scattering framework and it has been first analysed by Bethe Ansatz techniques in a famous paper by Lieb and Wu [117,118]. Coordinate Bethe Ansatz wave-functions for the Hubbard Hamiltonian eigenvectors have been obtained in [119] and subsequent papers. The quantum inverse scattering formulation has been achieved thanks to the Shastry's derivation of the $R$-matrix [120–122]. From this, the one-parameter family of commuting transfer matrices, generating the Hubbard Hamiltonian by standard logarithmic derivative, can be introduced. The proof that this $R$-matrix satisfies the Yang-Baxter equation has been given in [123]. In [124–126] a Nested Algebraic Bethe Ansatz for the Hubbard model has been introduced while the quantum transfer matrix approach to study the thermodynamics of the Hubbard model has been considered in [127]. Interestingly, the Hubbard Hamiltonian is invariant under the direct sum of two $Y(sl_2)$ Yangians, as derived

---

[1]See also [68,69,73–77] for some previous partial understating.

in [128–131], while the structure of the fusion relations for the Hubbard model have been studied in [132], see also [133] for the finite temperature case. It is also relevant to remark that under a strong coupling limit and some special choice of the remaining parameters [10], the Hubbard Hamiltonian leads to the Hamiltonian of the $t$-$Jgl_{1|2}$-supersymmetric model, another well known model for the description of the high-temperature superconductivity, see [134–136] and references therein.

While integrable quantum models naturally emerge in the description of 1+1 quantum or 2-dimensional statistical mechanics phenomena, they are not really confined to this realm. For example, they play a fundamental role in deriving exact results also for four-dimensional quantum field theory like the planar $N = 4$ Supersymmetric Yang-Mills (SYM) gauge theory, see the review paper [137] and references therein. In this context, integrability tools have been used to derive exact results, like characterizations of the scaling dimensions of local operators for general values of the coupling constant. Notably, such results can be used also as a test of the AdS/CFT correspondence in this planar limit. Indeed, holding for arbitrary values of the coupling, they allow for a verification of the agreement both at weak and strong couplings with the perturbative results obtained respectively in gauge and string theory contexts. Integrability is also becoming relevant in the exact computation of observables. Interestingly, quantum integrable higher rank spin and super-spin chains have found applications in the computation of correlation functions in $N = 4$ SYM, see *e.g.* [138–143]. The same integrable Hubbard model enters in the description of the planar $N = 4$ SYM gauge theory [137, 144] in the large volume asymptotic regime. Indeed, relevant examples are the connection between its dilatation generator at weak coupling and the Hubbard Hamiltonian derived in [145] and the equivalence shown[2] in [146, 147] between the bound state $S$-matrix for $AdS_5 \times S^5$ superstring [152–154] and two copies of the Shastry's $R$-matrix of the Hubbard model multiplied by a nontrivial dressing phase. Moreover, this $S$-matrix enjoys the Yangian symmetry associated to the centrally extended $su(2|2)$ superalgebra [152–155] and an Analytic Bethe Ansatz description of the spectrum has been introduced on this basis in [146, 147, 156].

The large spectrum of applications of these higher rank quantum integrable models and of the Hubbard model, clearly motivate our interest in their analysis by quantum separation of variables. Let us mention that the first interesting analysis toward the SoV description of higher rank models have been presented in [25, 28], see also [57]. More recently, in [157], by the exact analysis of quantum chains of small sizes, the spectrum of the Sklyanin's B-operator has been conjectured together with its diagonalizability for fundamental representations of $gl_3$ Yang-Baxter algebra associated to some classes of twisted boundary conditions. While in [158] the SoV basis has been constructed for non-compact representations. In [1, 159–161] we have solved the transfer matrix spectral problem[3] for a large class of higher rank quantum integrable models. That is for integrable quantum models associated to the fundamental representations of the $Y(gl_n)$ and $U_q(\widehat{gl_n})$ Yang-Baxter algebra and of the $Y(gl_n)$ reflection algebra. This has been done by introducing and developing a new SoV approach relying only on the integrable structure of the model, i.e. the commutative algebra of conserved charges. In [163, 164] our construction of SoV bases has been extended to general finite dimensional representations of $gl_n$ Yang-Baxter algebra with twisted boundary conditions. In [1], we have proven for the $gl_2$ representations and for small size $gl_3$ representations that our SoV bases can be made coinciding with the Sklyanin's ones, if Sklyanin's construction can be applied. In [163, 164] this statement has been extended to the higher rank cases, in this way providing an SoV proof[4]

---

[2]One should remark that the spin chain approaches, as those in [146,147], miss the so-called wrapping corrections of the AdS/CFT spectrum while a full description for this spectral problem has been proposed in [148] and thereafter extensively tested, see e.g. [149] and the reviews [150] and [151] for further developments.

[3]While in [162], we have described in detail how our approach works beyond fundamental representations for $Y(gl_2)$.

[4]Indeed, the first proof of this conjecture has been given in [165] in the nested Bethe Ansatz framework.

of the non-nested Bethe Ansatz representation conjectured in [157] of the transfer matrix eigenstates as Sklyanin's $B$-operator multiple action in the zeros of a polynomial $Q$-function on a given reference state.

Here, we construct the quantum Separation of Variables (SoV) bases in the representation spaces of both the fundamental inhomogeneous $gl_{\mathcal{M}|\mathcal{N}}$ Yang-Baxter superalgebras and the inhomogeneous Hubbard model under general quasi-periodic twisted boundary conditions defined by twist matrices having simple spectrum. Let us mention here an interesting proposal for a representation of the eigenvectors using a single $B$ operator in [166] for $gl_{\mathcal{M}|\mathcal{N}}$ models inspired by the SoV related methods [157]. In our approach, the SoV bases are constructed by using the known integrable structure of these quantum models, *i.e.* the associated commuting transfer matrices, following our general ideas introduced in [1]. The SoV bases are generated by the multiple actions of the transfer matrices on a generic co-vector of the Hilbert space. The fact that we are able to prove that such sets of co-vectors indeed form bases of the space of states implies important consequences on the spectrum of the transfer matrices. In fact, it follows that the transfer matrices have non degenerate (simple) spectrum or that they are diagonalizable with simple spectrum if the twist matrix respectively has simple spectrum or is diagonalizable with simple spectrum. Moreover, in our SoV bases the resolution of the transfer matrix eigenvalue problem is equivalent to the resolution of the full transfer matrix spectrum (eigenvalues and eigenvectors). Indeed, our SoV bases allow us to associate uniquely to any eigenvalue an eigenvector whose wave-function has the factorized form in terms of product of the transfer matrix eigenvalues on the spectrum of the separated variables.

It is worth pointing out that for these classes of higher rank quantum integrable models, fewer exact results are available when compared to those described for the best known examples of the XXZ spin 1/2 quantum integrable chains. Exact results are mainly confined to the spectral problem and only recently some breakthrough has been achieved toward the dynamics in the framework of the Nested Algebraic Bethe Ansatz (NABA) [167–170], for some higher rank spin and super-spin chains [165, 171–180].

More in detail, in the supersymmetric case, the associated spectral problem has been analysed by using the transfer matrix functional relations, generated by fusion [181–183] of irreducible representations in the auxiliary space of the representation. The Analytic Bethe Ansatz [183–185] developed in this functional framework has been applied to the spectral problem of these supersymmetric models. An important step in the systematic description and analysis of these functional equations has been done by rewriting them in Bazhanov and Reshetikhin's determinant form in [186], and in a Hirota bilinear difference equation form in [187–189]. These so-called $T$-systems appear both in classical and quantum integrability. An interesting account for their relevance and different application areas can be found in [190]. The validity of these fusion rules and of Analytic Bethe Ansatz description in the supersymmetric case have been derived in [191–194]. In [195, 196] a method has been developed and applied to the supersymmetric case based on the use of Bäcklund transformations on the Hirota-type functional equations [197–199]. It allows a systematic classification of the different Nested Algebraic Bethe Ansatz equations and $TQ$-functional equations, which emerge naturally in the supersymmetric case, due to different possible choices of the systems of simple roots. It also allows the identification of $QQ$-functional equations of Hirota type for the Baxter's $Q$-functions, see for example [200–202]. Nested Algebraic Bethe Ansatz [136, 170, 203] has been successfully used to get Bethe vectors representations for fundamental representations of $Y(gl_{\mathcal{M}|\mathcal{N}})$ and $U_q(\widehat{gl_{\mathcal{M}|\mathcal{N}}})$, see also the recent result [204], while determinant formulae for Bethe eigenvector norms, scalar products and some computations of form factors have been made accessible in [205–208] for the $Y(gl_{1|2})$ and $Y(gl_{2|1})$ case. The completeness of the Nested Algebraic Bethe Ansatz approach for supersymmetric Yangian representations

has been shown in the following papers[5] [209, 210], respectively for the representations of $Y(gl_{1|1})$ and of the general $Y(gl_{\mathcal{M}|\mathcal{N}})$, in the setup of the so-called $QQ$-Wronskian equations introduced in [211] for the non-supersymmetric case $Y(gl_{\mathcal{M}})$ and more recently in [204] for the $Y(gl_{\mathcal{M}|\mathcal{N}})$ models.

In this paper we start to develop the quantum separation of variables method for these supersymmetric integrable quantum models. The natural advantage of the SoV method is that it is not an Ansatz method and then the completeness of the spectrum description is mainly a built-in feature of it, as proven for a large class of quantum integrable models [39–41, 43–59]. More in detail, no Ansatz is done on the SoV representation of transfer matrix eigenvectors.[6] Indeed, their factorized wave-functions in terms of the eigenvalues of the transfer matrix, or of the Baxter's $Q$-operator [66, 212–233] are just a direct consequence of the form of the SoV basis. Moreover, these SoV representations are extremely simple and universal and should lead to determinant formulae for scalar products.[7] The SoV representation of transfer matrix eigenvectors also brings to Algebraic Bethe Ansatz rewriting of non-nested type for the eigenvectors,[8] *i.e.* as the action on a SoV induced "reference vector" of a single monomial of SoV induced "$B$-operators" over the zeros of an associated $Q$-operator eigenvalue [1]. It is worth to point out that this represents a strong simplification *w.r.t.* the eigenvector representation in NABA approach, where the holding different representations [170] are equivalent to an "explicit representation" which is written in the form of a sum over partitions. This type of results in the SoV framework is even more important in the case of the Hubbard model. Indeed, there, algebraic approaches like NABA are mainly limited to the two particle case [124] and other exact results are accessible only via coordinate Bethe Ansatz.[9]

The paper is organized as follows. In section 2, we first shortly present the graded formalism for the superalgebras $gl_{\mathcal{M}|\mathcal{N}}$ and their fundamental representations, we sum up the main properties of the hierarchy of the fused transfer matrices and their reconstruction in terms of the fundamental[10] one. The SoV basis is then constructed in subsection 2.4 by using the integrable structure of these models. In subsection 2.5, we make some general statement about the closure and admissibility conditions to fix the transfer matrix spectrum for these quantum integrable models. In section 3, we specialize the discussion on the $gl_{1|2}$ model. We state our conjecture on the corresponding closure conditions and we present some first arguments in favour of it in subsection 3.1. Then, we treat in detail a special twisted case in subsection 3.2, for which we prove that the entire spectrum of the transfer matrix is characterized by our conjecture. Then, we give a reformulation of the spectrum in terms of the solutions to a quantum spectral curve equation. Moreover, for these representations, we show the completeness of

---

[5]Both the papers [209, 210] appeared after the present paper and they are not directly related to our SoV approach.

[6]In the Bethe Ansatz framework, the fact that the form of the eigenvectors is fixed by the Ansatz implies that to prove the completeness of the spectrum description one has to define first admissibility conditions which generate nonzero vectors and then one has to count the number of these solutions and show that it coincides with the dimension of the representation space, in absence of Jordan blocks. This first step is for example done in the papers [204, 210, 211] through the introduction of the isomorphism to the $QQ$-Wronskian equations.

[7]Indeed, our recent results on higher rank scalar products [234] show the appearance of simple determinant formulae once the SoV basis are appropriately chosen, see also [235, 236] for some interesting SoV analysis of the higher rank scalar products.

[8]Note that this SoV versus ABA rewriting of transfer matrix eigenvectors was first observed in a rank 1 case in [30, 31] and it can be extended in general for polynomial $Q$-operators, as e.g. argued in [1]. One has to mention that these non-nested forms were first proposed in [157, 166] together with the form of the $B$-operator for rank higher than 2.

[9]In fact, to our knowledge, the generic N-particle transfer matrix eigenvalues are well verified guesses [10], no Bethe vectors representation is achieved for the corresponding eigenvectors and the conjectured norm formula [10] has to be proven yet.

[10]That is the transfer matrix associated to Lax operators on isomorphic auxiliary and quantum spaces, *i.e.* in our fundamental representations the transfer matrices with Lax operators coinciding with the $R$-matrix.

the Nested Algebraic Bethe Ansatz, by proving that any eigenvalue can be rewritten in a NABA form using our $Q$-functions. In section 4, we derive an SoV basis for the Hubbard model with general integrable twist matrix having simple spectrum. Finally, in appendix A for the $gl_{1|2}$ model with general integrable twist matrices, we verify that the NABA form of eigenvalues satisfies the closure and admissibility conditions, implying its compatibility with our conjecture in the SoV framework. In appendix B, we present the proof that our conjecture indeed completely characterizes the transfer matrix spectrum for any integrable twist matrix having simple spectrum for the model defined on two sites while we verify this property by numerical computations for three sites. In appendix C, we give a derivation of the closure relation for the $gl_{\mathcal{M}|\mathcal{N}}$ case.

# 2 Separation of variables for integrable $gl_{\mathcal{M}|\mathcal{N}}$ fundamental models

Graded structures and Lie superalgebras are treated in great details in [237, 238]. The quantum inverse scattering construction for graded models was introduced in [2–4], and summarized in many articles, see e.g. [10, 156, 239]. Details on Yangians structures for Lie superalgebras can be found in [240, 241].

For the article to be self contained, we introduce the graded algebra $gl_{\mathcal{M}|\mathcal{N}}$ and its fundamental Yangian model in the following, and make explicit the notations and rules for graded computations.

## 2.1 Graded formalism and integrable $gl_{\mathcal{M}|\mathcal{N}}$ fundamental models

A super vector space $V$ is a $\mathbb{Z}_2$-graded vector space, *ie.* we have

$$V = V_0 \oplus V_1. \tag{2.1}$$

Vectors of $V_0$ are even, while vectors of $V_1$ are odd. Objects that have a well-defined parity, either even or odd, are called homogeneous. The parity map, defined on homogeneous objects, writes

$$p : A \in V \longmapsto p(A) = \bar{A} = \begin{cases} 0 & \text{if } A \in V_0 \\ 1 & \text{if } A \in V_1 \end{cases}. \tag{2.2}$$

Maps between $\mathbb{Z}_2$-graded objects are called even if they preserve the parity of objects, or odd if they flip it. An associative superalgebra is a super vector space with an even multiplication map that is associative and the algebra has a unit element for the multiplication. For a superalgebra $V$ we have $V_i V_j \subseteq V_{i+j \pmod 2}$. A Lie superalgebra is a super vector space $\mathfrak{g} = \mathfrak{g}_0 \oplus \mathfrak{g}_1$ equipped with an even linear map $[\,,\,] : \mathfrak{g} \otimes \mathfrak{g} \to \mathfrak{g}$ that is graded antisymmetric and satisfies the graded Jacobi identity.

The set of linear maps from $V$ to itself is noted $\text{End } V$, and it is a $\mathbb{Z}_2$-graded vector space as well. It is an associative superalgebra with multiplication given by the composition. It is also a Lie superalgebra with the Lie super-bracket defined as the graded commutator between homogeneous objects for the multiplication

$$[A, B] = AB - (-1)^{\bar{A}\bar{B}} BA, \tag{2.3}$$

which extends linearly to the whole space. As a Lie superalgebra, it is denoted $gl(V)$.

**Tensor products**   The tensor product of two super vector spaces $V$ and $W$ is the tensor product of the underlying vectors spaces, with the $\mathbb{Z}_2$-grading structure given by

$$\text{for } k = 0 \text{ or } 1, \quad (V \otimes W)_k = \bigoplus_{i+j=k(\text{mod}\,2)} V_i \otimes W_j. \tag{2.4}$$

This also defines the tensor product of associative superalgebras, being defined on the underlying vector space structure, but then we have to define an associative multiplication compatible with the grading. For $A, B$ two associative superalgebras, the multiplication rule on $A \otimes B$ is given by

$$(a_1 \otimes b_1)(a_2 \otimes b_2) = (-1)^{\bar{b}_1 \bar{a}_2} a_1 a_2 \otimes b_1 b_2, \tag{2.5}$$

for $a_1, a_2 \in A$ and $b_1, b_2 \in B$ homogeneous, and extends linearly to $A \otimes B$.

This rule of sign also appears in the action of $A \otimes B$ on $V \otimes W$, where $V$ is an $A$-module and $W$ is a $B$-module. We have

$$(a \otimes b) \cdot (v \otimes w) = (-1)^{\bar{b} \bar{v}} a \cdot v \otimes b \cdot w, \tag{2.6}$$

for $a \in A$, $b \in B$, $v \in V$ and $w \in W$.

This rule extends naturally to $N$-fold tensor product, $N > 2$. For example,

$$(a_1 \otimes b_1 \otimes c_1)(a_2 \otimes b_2 \otimes c_2) = (-1)^{\bar{a}_2(\bar{b}_1 + \bar{c}_1) + \bar{b}_2 \bar{c}_1} a_1 a_2 \otimes b_1 b_2 \otimes c_1 c_2. \tag{2.7}$$

For some authors, the above construction goes explicitly by the name of *super* or *graded* tensor product. We will stick to the name tensor product.[11]

**Lie superalgebra** $gl_{\mathcal{M}|\mathcal{N}}$   Let $V = \mathbb{C}^{\mathcal{M}|\mathcal{N}}$ be the complex vector superspace with even part of dimension $\mathcal{M}$ and odd part of dimension $\mathcal{N}$. The general linear Lie algebra $gl_{\mathcal{M}|\mathcal{N}} = gl(\mathbb{C}^{\mathcal{M}|\mathcal{N}})$ is the $\mathbb{Z}_2$-graded vector space $\text{End}\,\mathbb{C}^{\mathcal{M}|\mathcal{N}}$ with the Lie super-bracket defined by the graded commutator (2.3).

We fix a homogeneous basis $\{v_1, \ldots, v_{\mathcal{M}}, v_{\mathcal{M}+1}, \ldots, v_{\mathcal{M}+\mathcal{N}}\}$ of $\mathbb{C}^{\mathcal{M}|\mathcal{N}}$, where $v_i$ is even for $i \leq \mathcal{M}$ and odd for $i \geq \mathcal{M}+1$ and we assign a parity to the index themselves for convenience: $\bar{i} = 0$ for $i \leq \mathcal{M}$ and $\bar{i} = 1$ for $i \geq \mathcal{M}+1$.

The elementary operators $e_i^j$ of $gl_{\mathcal{M}|\mathcal{N}}$ have parity $p(e_i^j) = \bar{i} + \bar{j}(\text{mod}\,2)$. They are defined by their action on the basis of $V$ by

$$e_i^j \cdot v_k = \delta_k^j v_i. \tag{2.8}$$

Since they multiply as

$$e_i^j e_k^l = \delta_k^j e_i^l, \tag{2.9}$$

it follows that the graded commutator is

$$\left[ e_i^j, e_k^l \right] = e_i^j e_k^l - (-1)^{p(e_i^j)p(e_k^l)} e_k^l e_i^j = \delta_k^j e_i^l - (-1)^{(\bar{i}+\bar{j})(\bar{k}+\bar{l})} \delta_i^l e_k^j. \tag{2.10}$$

Elements of $gl_{\mathcal{M}|\mathcal{N}}$ decompose on the elementary operators as

$$a = \sum_{i,j=1}^{\mathcal{M}+\mathcal{N}} a_j^i e_i^j \equiv a_j^i e_i^j, \tag{2.11}$$

---

[11]Note also that some authors prefer to use a matrix formalism and by "super tensor product" denote a morphism between graded and non-graded structure, see [4], appendix A of [136] or appendix A [195]. We will not make any extensive use of it in the following.

where in the last term the sum over repeated indexes is omitted, as we will do in the following. Elements of $\left(gl_{\mathcal{M}|\mathcal{N}}\right)^{\otimes N}$ writes

$$A = A^{i_1 \dots i_N}_{j_1 \dots j_N} \, e^{j_1}_{i_1} \otimes \dots \otimes e^{j_N}_{i_N}. \tag{2.12}$$

Note that due to the sign rule (2.5), the coordinates $A^{i_1 \dots i_N}_{j_1 \dots j_N}$ do not coincide with the components of the image of $v_{j_1} \otimes \dots \otimes v_{j_N}$ in the tensored basis of $V^{\otimes N}$:

$$A \cdot v_{j_1} \otimes \dots \otimes v_{j_N} = (-1)^{\sum_{k=1}^{N-1} \bar{i}_k (\bar{i}_{k+1} + \dots + \bar{i}_N)} A^{i_1 \dots i_N}_{j_1 \dots j_N} \, v_{i_1} \otimes \dots \otimes v_{i_N}. \tag{2.13}$$

In the non-graded case, these two tensors would be identical.

One may use the coordinates expression to check the parity of a given operator of $\left(gl_{\mathcal{M}|\mathcal{N}}\right)^{\otimes N}$. An operator $A$ is homogeneous of parity $p(A)$ if

$$\forall i_1, \dots, i_N, j_1, \dots, j_N, \quad (-1)^{\bar{i}_1 + \bar{j}_1 + \dots + \bar{i}_N + \bar{j}_N} A^{i_1 \dots i_N}_{j_1 \dots j_N} = (-1)^{p(A)} A^{i_1 \dots i_N}_{j_1 \dots j_N}. \tag{2.14}$$

The supertrace is defined on the elementary operators as $\operatorname{str} e^j_i = (-1)^{\bar{j}} \delta^j_i$. Elements of $gl_{\mathcal{M}|\mathcal{N}}$ may write as a block matrix

$$A = \begin{pmatrix} A_{(\mathcal{M},\mathcal{M})} & A_{(\mathcal{M},\mathcal{N})} \\ A_{(\mathcal{N},\mathcal{M})} & A_{(\mathcal{N},\mathcal{N})} \end{pmatrix} \in gl_{\mathcal{M}|\mathcal{N}}, \tag{2.15}$$

where $A_{(\mathcal{M},\mathcal{M})}$ is an $\mathcal{M}$ by $\mathcal{M}$ square matrix, $A_{(\mathcal{M},\mathcal{N})}$ an $\mathcal{M}$ by $\mathcal{N}$ square matrix, etc. Hence we have $\operatorname{str} A = \operatorname{tr} A_{(\mathcal{M},\mathcal{M})} - \operatorname{tr} A_{(\mathcal{N},\mathcal{N})}$. Note that the supertrace vanishes on the graded commutator

$$\operatorname{str}([A, B]) = 0. \tag{2.16}$$

**Dual space**   Let us denote $|i\rangle \equiv v_i$. The dual basis $\{\langle j|\}_{j=1,\dots,\mathcal{M}+\mathcal{N}}$ is defined by

$$\forall i, \quad \langle j|i\rangle = \delta_{ji}. \tag{2.17}$$

The covectors are graded by $p(\langle j|) = \bar{j}$. The dual of a vector $|\psi\rangle = \psi_i |i\rangle$ is

$$\langle \psi| = (\psi_i |i\rangle)^\dagger = \psi_i^* \langle i|, \tag{2.18}$$

where the star $*$ stands for the complex conjugation. For $V^{\otimes N}$, the dual basis covectors have an additional sign in their definition

$$(|i_1\rangle \otimes \dots \otimes |i_N\rangle)^\dagger \equiv \langle i_1| \otimes \dots \otimes \langle i_N| (-1)^{\sum_{k=2}^N \bar{i}_k (\bar{i}_1 + \dots + \bar{i}_{k-1})}, \tag{2.19}$$

such that it compensates for the permutation of vectors and covectors:

$$\begin{aligned}(|i_1\rangle \otimes \dots \otimes |i_N\rangle)^\dagger (|j_1\rangle \otimes \dots \otimes |j_N\rangle) &= (-1)^{\sum_{k=2}^N \bar{i}_k (\bar{i}_1 + \dots + \bar{i}_{k+1}) + \sum_{k=2}^N \bar{i}_k (\bar{j}_1 + \dots + \bar{j}_{k-1})} \langle i_1|j_1\rangle \dots \langle i_N|j_N\rangle \\ &= \delta_{i_1 j_1} \dots \delta_{i_N j_N}.\end{aligned} \tag{2.20}$$

Similarly, for N even operators $A_1, \dots, A_N$ where each $A_j$ acts non-trivially only in the $j^{\text{th}}$ space of the tensor product $V^{\otimes N}$, the following matrix element factorizes over the tensorands

$$(|i_1\rangle \otimes \dots \otimes |i_N\rangle)^\dagger A_1 \dots A_N (|j_1\rangle \otimes \dots \otimes |j_N\rangle) = \langle i_1|A_1|j_1\rangle \dots \langle i_N|A_N|j_N\rangle. \tag{2.21}$$

Indeed, through the matrix elements $\langle i_a|K_a|j_a\rangle$ that arise from the calculation, the evenness of $K$ forces the grading $\bar{i}_a$ and $\bar{j}_a$ at site $a$ to be equal, so the signs compensate.

**Permutation operator** The permutation operator has to take account of the grading when flipping vectors $\mathbb{P} \cdot (v \otimes w) = (-1)^{\bar{v}\bar{w}} w \otimes v$. Thus we have

$$\mathbb{P} = (-1)^{\bar{\beta}} e_\alpha^\beta \otimes e_\beta^\alpha, \tag{2.22}$$

$$\mathbb{P} \cdot (v_i \otimes v_j) = (-1)^{\bar{i}\bar{j}} v_j \otimes v_i. \tag{2.23}$$

Remark the additional signs in the action as discussed in (2.13). For two homogeneous operators $A$ and $B$,

$$\mathbb{P}(A \otimes B)\mathbb{P} = (-1)^{p(A)p(B)} B \otimes A. \tag{2.24}$$

On an N-fold tensor product $V_1 \otimes \ldots \otimes V_N$, with $V_i \simeq \mathbb{C}^{\mathcal{M}|\mathcal{N}}$, the permutation operator $P_{ab}$ between spaces $V_a$ and $V_b$ writes

$$\mathbb{P}_{ab} = (-1)^{\bar{\beta}} \, \mathbb{I} \otimes \ldots \otimes \mathbb{I} \otimes \underbrace{e_\alpha^\beta}_{\text{site } a} \otimes \mathbb{I} \ldots \otimes \mathbb{I} \otimes \underbrace{e_\beta^\alpha}_{\text{site } b} \otimes \mathbb{I} \ldots \otimes \mathbb{I}, \tag{2.25}$$

where the number of identity operators is obvious by the context. The permutation operator is globally even. We have $\mathbb{P}_{ab}^2 = \mathbb{I}^{\otimes N}$, and the usual identities are verified

$$\mathbb{P}_{12} = \mathbb{P}_{21}, \tag{2.26}$$

$$\mathbb{P}_{12}\mathbb{P}_{13} = \mathbb{P}_{13}\mathbb{P}_{23} = \mathbb{P}_{23}\mathbb{P}_{12}, \tag{2.27}$$

$$\mathbb{P}_{13}\mathbb{P}_{24} = \mathbb{P}_{24}\mathbb{P}_{13}, \quad \mathbb{P}_{12}\mathbb{P}_{34} = \mathbb{P}_{34}\mathbb{P}_{12}, \tag{2.28}$$

which extend naturally to a N-fold tensor product.

**The $\mathscr{Y}(gl_{\mathcal{M}|\mathcal{N}})$ fundamental model** The $R$ matrix for the fundamental model of the Yangian $\mathscr{Y}(gl_{\mathcal{M}|\mathcal{N}})$ writes

$$R(\lambda, \mu) = (\lambda - \mu)\mathbb{I} \otimes \mathbb{I} + \eta \mathbb{P} \quad \in \text{End}(\mathbb{C}^{\mathcal{M}|\mathcal{N}} \otimes \mathbb{C}^{\mathcal{M}|\mathcal{N}}). \tag{2.29}$$

It is of difference type and decomposes on elementary operators as $R(\lambda) = R_{jl}^{ik}(\lambda) e_i^j \otimes e_k^l$ with

$$R_{jl}^{ik}(\lambda) \equiv \lambda \, \delta_j^i \delta_l^k + \eta \, (-1)^{\bar{j}} \delta_l^i \delta_j^k. \tag{2.30}$$

It generalizes to a N-fold tensor product : the matrix $R_{ab}(\lambda) = \lambda \mathbb{I}^{\otimes(N+1)} + \eta \mathbb{P}_{ab}$ of $\text{End}(V_1 \otimes \ldots \otimes V_N)$ who acts non trivially only on $V_a$ and $V_b$ writes

$$R_{ab}(\lambda) = R_{jl}^{ik}(\lambda) \, \mathbb{I} \otimes \ldots \otimes \mathbb{I} \otimes \underbrace{e_i^j}_{\text{site } a} \otimes \mathbb{I} \ldots \otimes \mathbb{I} \otimes \underbrace{e_k^l}_{\text{site } b} \otimes \mathbb{I} \ldots \otimes \mathbb{I}, \tag{2.31}$$

using the generic notation (2.30). The $R$ matrix is globally even and satisfies the Yang-Baxter equation

$$R_{12}(\lambda - \mu)R_{13}(\lambda)R_{23}(\mu) = R_{23}(\mu)R_{13}(\lambda)R_{12}(\lambda - \mu). \tag{2.32}$$

Sometimes the Yang-Baxter equation is written in coordinates and is explicitly referred to as a "graded" version [2, 3]. By equation (2.13),

$$R(\lambda)v_j \otimes v_l = \mathsf{R}_{jl}^{ik}(\lambda)v_i \otimes v_k, \tag{2.33}$$

with $\mathsf{R}_{jl}^{ik}(\lambda) = \lambda\delta_j^i\delta_l^k + \eta(-1)^{\bar{i}\bar{k}}\delta_l^i\delta_j^k$. Then we have

$$\mathsf{R}_{\alpha'\beta'}^{\alpha\beta}(\lambda,\mu)\mathsf{R}_{\alpha''\gamma'}^{\alpha'\gamma}(\lambda)\mathsf{R}_{\beta''\gamma''}^{\beta'\gamma'}(\mu)(-1)^{\bar{\beta}'(\bar{\alpha}'+\bar{\alpha}'')} = \mathsf{R}_{\beta'\gamma'}^{\beta\gamma}(\mu)\mathsf{R}_{\alpha'\gamma''}^{\alpha\gamma'}(\lambda)\mathsf{R}_{\alpha''\beta''}^{\alpha'\beta'}(\lambda-\mu)(-1)^{\bar{\beta}'(\bar{\alpha}+\bar{\alpha}')}. \tag{2.34}$$

One may check the $gl_{\mathcal{M}|\mathcal{N}}$ invariance of the $R$ matrix (2.29)

$$\forall x \in gl_{\mathcal{M}|\mathcal{N}}, \quad [R(\lambda), x \otimes \mathbb{I} + \mathbb{I} \otimes x] = 0. \tag{2.35}$$

If $K$ is an homogeneous even matrix of $gl_{\mathcal{M}|\mathcal{N}}$ of the form

$$K = \begin{pmatrix} K_{\mathcal{M}} & 0 \\ 0 & K_{\mathcal{N}} \end{pmatrix}, \tag{2.36}$$

we have

$$R(\lambda)(K \otimes \mathbb{I})(\mathbb{I} \otimes K) = (\mathbb{I} \otimes K)(K \otimes \mathbb{I})R(\lambda). \tag{2.37}$$

This is a scalar version of the Yang-Baxter equation (2.32), where we put a trivial representation on the third space.

For a spin chain of length N, we denote the Hilbert space by $\mathcal{H} = V_1 \otimes \ldots \otimes V_N$ and the auxiliary space by $V_0$, all the $V_j$ superspaces being isomorphic to $\mathbb{C}^{\mathcal{M}|\mathcal{N}}$. Taking an even twist (2.36), the monodromy is an element of $\mathrm{End}(V_0 \otimes V_1 \otimes \ldots \otimes V_N)$ and writes

$$M_0^{(K)}(\lambda) = K_0 R_{0N}(\lambda - \xi_N) \ldots R_{01}(\lambda - \xi_1), \tag{2.38}$$

where $\xi_1, \ldots, \xi_N$ are the inhomogeneities of the chain. The monodromy is globally even as a product and tensor product of even operators. In coordinates, using the notation $R_{jl}^{ik}$ of (2.30), it writes

$$M^{(K)}(\lambda) = M_{j}^{i}{}_{\beta_1 \ldots \beta_N}^{\alpha_1 \ldots \alpha_N}(\lambda) \, e_i^j \otimes e_{\alpha_1}^{\beta_1} \otimes \ldots \otimes e_{\alpha_N}^{\beta_N} \tag{2.39}$$

$$= K_{j_N}^{i} R_{j_{N-1}\beta_N}^{j_N \alpha_N}(\lambda - \xi_N) \ldots R_{j\beta_1}^{j_1 \alpha_1}(\lambda - \xi_1) \, e_i^j \otimes e_{\alpha_1}^{\beta_1} \otimes \ldots \otimes e_{\alpha_N}^{\beta_N}, \tag{2.40}$$

where all the signs from the multiplication of the operators actually vanish because of the evenness of $R$. We are dropping the superscript $(K)$ from the coordinates to make the notation less cluttered. Writing $M^{(K)}(\lambda) = e_i^j \otimes M_j^i(\lambda)$, the above expression shows that the monodromy elements $M_j^i(\lambda) \in \mathrm{End}(\mathcal{H})$ are homogeneous of parity $p(M_j^i(\lambda)) = \bar{i} + \bar{j}$.

The Yang-Baxter scheme generalizes to the monodromy thanks to the global evenness of the $R$ matrix and the form (2.36) of the twist, and we have

$$R_{ab}(\lambda - \mu) M_a^{(K)}(\lambda) M_b^{(K)}(\mu) = M_b^{(K)}(\mu) M_a^{(K)}(\lambda) R_{ab}(\lambda - \mu). \tag{2.41}$$

One can prove the $\mathcal{Y}(gl_{\mathcal{M}|\mathcal{N}})$ Yang-Baxter relations between the monodromy elements write as

$$\left[ M_i^j(\lambda), M_k^l(\mu) \right] = (-1)^{\bar{i}\bar{k} + \bar{i}\bar{l} + \bar{k}\bar{l}} \left( M_k^j(\mu) M_i^l(\lambda) - M_k^j(\lambda) M_i^l(\mu) \right), \tag{2.42}$$

where $\left[ M_i^j(\lambda), M_k^l(\mu) \right]$ is the graded commutator (2.3).

The transfer matrix is obtained by taking the supertrace over the auxiliary space $V_0$

$$T^{(K)}(\lambda) = \mathrm{str}_0 \, M_0^{(K)}(\lambda). \tag{2.43}$$

It is an even operator of $\mathrm{End}(\mathcal{H})$ as a sum of diagonal elements of the monodromy. Because the supertrace vanishes on the graded commutator, one proves the commutation of the transfer matrices

$$\forall \lambda, \mu \in \mathbb{C}, \quad \left[ T^{(K)}(\lambda), T^{(K)}(\mu) \right] = 0. \tag{2.44}$$

## 2.2 The tower of fused transfer matrices

Tensor products of fundamental representations of $gl_{\mathcal{M}|\mathcal{N}}$ decompose in direct sum of irreducible subrepresentations (irreps). Young diagrams are used to carry out calculations with a mechanic proper to superalgebras, though very similar to the non graded case [195,242–244]. Finite dimensional irreducible representations are labelled in a unique way by Kac-Dynkin labels, but the correspondence between Kac-Dynkin labels and Young diagrams is not one-to-one [245].

Admissible Young diagrams lie inside a *fat hook* domain pictured in figure 1, defined in the $(a, b)$ bidimensional lattice as $H_{\mathcal{M}|\mathcal{N}} \equiv (\mathbb{Z}_{\geq 1} \times \mathbb{Z}_{\geq 1}) \setminus (\mathbb{Z}_{>\mathcal{M}} \times \mathbb{Z}_{>\mathcal{N}})$. Young diagrams can expand infinitely in both $a$ and $b$ directions, but the box $(a \geq \mathcal{N}+1, b \geq \mathcal{M}+1)$ is forbidden, leading to the hook shape.

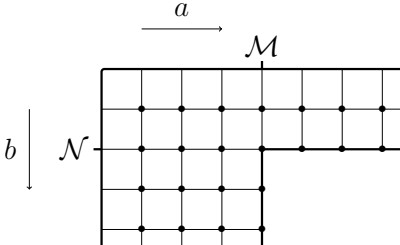

Figure 1: The fat hook domain $H_{\mathcal{M}|\mathcal{N}}$ of admissible Young diagrams for the superalgebra $gl_{\mathcal{M}|\mathcal{N}}$. Bullet points correspond to admissible coordinates $(a, b)$ defining rectangular Young diagrams.

*Remark* 2.1. For $\mathcal{N} = 0$, the fat hook degenerates to a vertical strip, forcing $a \leq \mathcal{M}$. We recover the usual Young diagram indexation of $gl(\mathcal{M})$ irreducible representations, though the diagrams are here displayed vertically, corresponding to the transposition of the usual $gl(\mathcal{M})$ ones. This is consistent, for example, with the convention of [195], if one rotate the diagrams found there by $-\pi/2$.

The tensoring procedure is called fusion in the context of the quantum inverse scattering method [181,182]. It is used to generate higher dimensional $gl_{\mathcal{M}|\mathcal{N}}$-invariant $R$-matrices. The Yang-Baxter scheme is preserved, as degeneracy points of the fundamental $R$-matrix allow to construct the projectors $P_\lambda : (\mathbb{C}^{\mathcal{M}|\mathcal{N}})^{\otimes n} \to V_\lambda$, that extract the wanted subrepresentation $\lambda$, as a product of $R$-matrices. Fusing on the auxiliary space from $M_0^{(K)}$, we obtain new monodromy operators and thus new transfer matrices of $\mathrm{End}(\mathcal{H})$.

For a rectangular Young tableau corresponding to the point $(a, b) \in H_{\mathcal{M}|\mathcal{N}}$, with $b$ rows and $a$ columns, the monodromy matrix reads

$$M_b^{(a),(K)}(\lambda) \equiv P_b^{(a)} \left[ \overleftarrow{\bigotimes}_{\substack{1 \leq s \leq a \\ 1 \leq r \leq b}} M^{(K)}(\lambda + \eta(r-s)) \right] P_b^{(a)}, \tag{2.45}$$

and the transfer matrix

$$T_b^{(a),(K)}(u) \equiv \mathrm{str}_{V_b^{(a)}} M_b^{(a),(K)}(u) \tag{2.46}$$

is obtained by taking the supertrace over the $a \times b$ auxiliary spaces $V \simeq \mathbb{C}^{\mathcal{M}+\mathcal{N}}$, with $V_b^{(a)} \equiv V \otimes \ldots \otimes V$, $ab$ times.

The shifts in (2.45) are given by filling the fat hook as in figure 2. We then read the rectangular Young diagram column by column, top to bottom from left to right, and tensor the shifted monodromy (2.38) corresponding to the current box to the left of the previous ones.

| 0 | $-\eta$ | $-2\eta$ | $-3\eta$ | $-4\eta$ |
|---|---|---|---|---|
| $\eta$ | 0 | $-\eta$ | $-2\eta$ | $-3\eta$ |
| $2\eta$ | $\eta$ | 0 | | |
| $3\eta$ | $2\eta$ | $\eta$ | | |

Figure 2: The domain $H_{\mathcal{M}|\mathcal{N}}$ is filled with multiples of the deformation parameter $\eta$. Starting from 0 in box $(1,1)$, we add $\eta$ when moving down and $-\eta$ when going right.

As said earlier, the projectors $P_\lambda$ can be constructed as a product of $R$-matrices, like in the non graded case [181, 195, 199]. For rectangular diagrams $(a, b)$, we have[12]

$$P_b^{(a)} \propto \prod_{i<j} R_{ij}(s_j - s_i), \tag{2.47}$$

where $i, j$ run on the boxes of the diagram of figure 2 column by column, top to bottom from left to right, and $s_i, s_j$ are the shift contained in the boxes.

All these transfer matrices commute with each other, as consequence of the Yang-Baxter equation (2.32) being true for any irreps taken in the spaces 1, 2, and 3

$$\forall (a, b), (c, d) \in H_{\mathcal{M}|\mathcal{N}}, \forall \lambda, \mu \in \mathbb{C}, \quad \left[ T_b^{(a),(K)}(\lambda), T_d^{(c),(K)}(\mu) \right] = 0. \tag{2.48}$$

Let us comment that through the nice coderivative formalism [246, 247], in a different but equivalent manner, these fused transfer matrices and the following fusion properties can be also derived. In particular, we make use of this formalism in appendix C to verify (2.54).

Among many others, the fused transfer matrices satisfy the following important properties [191–195]:

**Polynomial structure** The generic fused transfer matrix $T_b^{(a),(K)}(\lambda)$ is polynomial in $\lambda$ of degree $ab\mathsf{N}$, with $(ab-1)\mathsf{N}$ central zeros given by

$$Z_b^{(a)}(\lambda) \equiv \prod_{n=1}^{\mathsf{N}} \left[ (\lambda - \xi_n)^{-1} \prod_{l=1}^{b} \prod_{m=1}^{a} (\lambda - \xi_n + \eta(l - m)) \right], \tag{2.49}$$

and therefore factorizes as

$$T_b^{(a),(K)}(\lambda) = \widetilde{T}_b^{(a),(K)}(\lambda) Z_b^{(a)}(\lambda), \tag{2.50}$$

where $\widetilde{T}_b^{(a),(K)}(\lambda)$ is polynomial in $\lambda$ of degree $\mathsf{N}$.

**Fusion equations** There is a bilinear relation between transfer matrices associated to adjacent rectangular diagrams

$$T_b^{(a),(K)}(\lambda - \eta) T_b^{(a),(K)}(\lambda) = T_{b+1}^{(a),(K)}(\lambda - \eta) T_{b-1}^{(a),(K)}(\lambda) + T_b^{(a-1),(K)}(\lambda - \eta) T_b^{(a+1),(K)}(\lambda), \tag{2.51}$$

where in our normalization, the following boundary conditions are imposed for consistency

$$T_{b\geq 1}^{(0),(K)}(\lambda) = T_0^{(a\geq 1),(K)}(\lambda) = 1. \tag{2.52}$$

---

[12]Note that in fact one has to use Cherednik regularization to extract the wanted projectors when the diagrams are not purely of row or column type [199].

All the fused transfer matrices outside the *extended fat hook* $\bar{H}_{\mathcal{M}|\mathcal{N}} \equiv (\mathbb{Z}_{\geq 0} \times \mathbb{Z}_{\geq 0})/(\mathbb{Z}_{>\mathcal{M}} \times \mathbb{Z}_{>\mathcal{N}})$ are identically zero, i.e.

$$\forall (a, b) \notin \bar{H}_{\mathcal{M}|\mathcal{N}}, \quad T_b^{(a),(K)}(\lambda) = 0. \tag{2.53}$$

These relations come from the Jacobi identity applied on the determinant form of the transfer matrices given by the Bazhanov-Reshetikhin formula (2.57), (2.58). We exclude the case $(a, b) = (0, 0)$ from the system of the $T_b^{(a)}(\lambda)$, as it cannot be defined uniquely from the boundary conditions.

**Inner-boundary condition**    As the correspondence between Young diagrams and irreps is not bijective, there exist non-trivial relations linking transfer matrices coming from distinct Young diagrams. This is especially the case for rectangular diagrams saturating one of the branch of the fat hook $\bar{H}_{\mathcal{M}|\mathcal{N}}$ [195,196,245]. We shall call the first of these relation the *inner-boundary* condition, which writes

$$(-1)^{\mathcal{N}} \text{Ber}(\lambda) T_{\mathcal{N}}^{(\mathcal{M}+1),(K)}(\lambda + \eta) = T_{\mathcal{N}+1}^{(\mathcal{M}),(K)}(\lambda), \tag{2.54}$$

where

$$\text{Ber}(\lambda) = \frac{\det K_{\mathcal{M}}}{\det K_{\mathcal{N}}} \frac{a(\lambda) \prod_{k=1}^{\mathcal{M}-1} d(\lambda - k\eta)}{\prod_{l=1-\mathcal{M}}^{\mathcal{N}-\mathcal{M}} d(\lambda + l\eta)} \mathbb{I}_{\mathscr{H}}, \tag{2.55}$$

and

$$a(\lambda - \eta) = d(\lambda) \equiv \prod_{n=1}^{\mathsf{N}} (\lambda - \xi_n). \tag{2.56}$$

$\text{Ber}(\lambda)$ coincides with the central element called the quantum Berezinian, for $\mathcal{N} \neq \mathcal{M}$, and it plays a role similar to the quantum determinant in the non graded case [156,240]. As anticipated, we verify this relation in appendix C.

## 2.3   Reconstruction of fused transfer matrix in terms of the fundamental one

Here we want to recall that all these fused transfer matrices are completely determined in terms of the transfer matrix $T_1^{(K)}(\lambda)$ obtained in (2.43). Indeed, the Bazhanov and Reshetikhin's determinant formulae [186] allows us to write all the $T_b^{(a),(K)}(\lambda)$ in terms of those of column type $T_r^{(1),(K)}(\lambda)$ and row type $T_1^{(r),(K)}(\lambda)$ by:

$$T_b^{(a),(K)}(\lambda) = \det_{1 \leq i,j \leq a} T_{b+i-j}^{(1),(K)}(\lambda - (i-1)\eta) \tag{2.57}$$

$$= \det_{1 \leq i,j \leq b} T_1^{(a+i-j),(K)}(\lambda + (i-1)\eta), \tag{2.58}$$

then our statement is proven once we prove it for the transfer matrix of type $T_r^{(1),(K)}(\lambda)$ and $T_1^{(r),(K)}(\lambda)$. Let us use in the following the simpler notations

$$T_a^{(K)}(\lambda) \equiv T_a^{(1),(K)}(\lambda), \quad T_{(a)}^{(K)}(\lambda) \equiv T_1^{(a),(K)}(\lambda), \tag{2.59}$$

and similarly

$$\widetilde{T}_a^{(K)}(\lambda) \equiv \widetilde{T}_a^{(1),(K)}(\lambda), \quad \widetilde{T}_{(a)}^{(K)}(\lambda) \equiv \widetilde{T}_1^{(a),(K)}(\lambda). \tag{2.60}$$

The fused transfer matrices $T_a^{(K)}(\lambda)$ and $T_{(a)}^{(K)}(\lambda)$ are polynomials of degree $a\mathsf{N}$ in $\lambda$, while $\widetilde{T}_a^{(K)}(\lambda)$ and $\widetilde{T}_{(a)}^{(K)}(\lambda)$ are of degree $\mathsf{N}$. We have the following properties for these matrices:

**Lemma 2.1.** *The following asymptotics hold:*

$$T_{\infty,a}^{(K)} \equiv \lim_{\lambda \to \infty} \lambda^{-aN} T_a^{(K)}(\lambda) = \text{str}_{1\dots a} P_{1\dots a}^+ K_1 \dots K_a P_{1\dots a}^+, \tag{2.61}$$

$$T_{\infty,(a)}^{(K)} \equiv \lim_{\lambda \to \infty} \lambda^{-aN} T_{(a)}^{(K)}(\lambda) = \text{str}_{1\dots a} P_{1\dots a}^- K_1 \dots K_a P_{1\dots a}^-, \tag{2.62}$$

*where, in agreement with (2.47), the projectors admit the following iterative representations in terms of the R-matrix:*

$$P_a^{(1)} \equiv P_{1\dots a}^+ = \frac{1}{a\eta} P_{1\dots a-1}^+ R((a-1)\eta) P_{2\dots a}^+, \tag{2.63}$$

$$P_1^{(a)} \equiv P_{1\dots a}^- = -\frac{1}{a\eta} P_{1\dots a-1}^- R(-(a-1)\eta) P_{2\dots a}^-. \tag{2.64}$$

*In the inhomogeneities the following fusion relations hold:*

$$T_{n+1}^{(K)}(\xi_a) = T_1^{(K)}(\xi_a) T_n^{(K)}(\xi_a + \eta), \tag{2.65}$$

$$T_{(n+1)}^{(K)}(\xi_a) = T_1^{(K)}(\xi_a) T_{(n)}^{(K)}(\xi_a - \eta), \tag{2.66}$$

*for any positive integer n.*

*Proof.* The asymptotics are an easy corollary of the definition of the fused transfer matrices of type $T_r^{(K)}(\lambda)$ and $T_{(r)}^{(K)}(\lambda)$. Let us now prove the fusion relations in the inhomogeneities, for $n = 1$ the identity:

$$T_2^{(K)}(\xi_a) = T_1^{(K)}(\xi_a) T_1^{(K)}(\xi_a + \eta), \tag{2.67}$$

$$T_{(2)}^{(K)}(\xi_a) = T_1^{(K)}(\xi_a) T_1^{(K)}(\xi_a - \eta), \tag{2.68}$$

are obtained by the fusion equations (2.51) just remarking that being:

$$Z_1^{(2)}(\lambda) = d(\lambda - \eta), \quad Z_2^{(1)}(\lambda) = d(\lambda + \eta), \tag{2.69}$$

it holds

$$T_{(2)}^{(K)}(\xi_a + \eta) = 0, \quad T_2^{(K)}(\xi_a - \eta) = 0. \tag{2.70}$$

Then we can proceed by induction to prove the identity, let us assume that it holds for $n \geq 1$ and let us prove it for $n + 1$, the relevant fusion identities reads:

$$\begin{aligned} T_n^{(K)}(\xi_a + \eta) T_n^{(K)}(\xi_a) = \; & T_{n+1}^{(K)}(\xi_a) T_{n-1}^{(K)}(\xi_a + \eta) \\ & + T_n^{(0),(K)}(\xi_a) T_n^{(2),(K)}(\xi_a + \eta), \end{aligned} \tag{2.71}$$

and

$$\begin{aligned} T_{(n)}^{(K)}(\xi_a - \eta) T_{(n)}^{(K)}(\xi_a) = \; & T_2^{(n),(K)}(\xi_a - \eta) T_0^{(n),(K)}(\xi_a) \\ & + T_{(n-1)}^{(K)}(\xi_a - \eta) T_{(n+1)}^{(K)}(\xi_a), \end{aligned} \tag{2.72}$$

which being:

$$Z_n^{(2)}(\lambda) \propto d(\lambda - \eta), \quad Z_2^{(n)}(\lambda) \propto d(\lambda + \eta), \tag{2.73}$$

read:

$$T_n^{(K)}(\xi_a + \eta) T_n^{(K)}(\xi_a) = T_{n+1}^{(K)}(\xi_a) T_{n-1}^{(K)}(\xi_a + \eta), \tag{2.74}$$

and

$$T_{(n)}^{(K)}(\xi_a - \eta)T_{(n)}^{(K)}(\xi_a) = T_{(n-1)}^{(K)}(\xi_a - \eta)T_{(n+1)}^{(K)}(\xi_a), \tag{2.75}$$

which lead to our identities for $n + 1$ once we use the induction hypothesis for $n$

$$T_n^{(K)}(\xi_a) = T_1^{(K)}(\xi_a)T_{n-1}^{(K)}(\xi_a + \eta), \tag{2.76}$$

$$T_{(n)}^{(K)}(\xi_a) = T_1^{(K)}(\xi_a)T_{(n-1)}^{(K)}(\xi_a - \eta). \tag{2.77}$$

□

The known central zeros and asymptotic imply that the interpolation formula in the $N$ special points defined by the fusion equations to write $T_n^{(K)}(\xi_a)$ and $T_{(n)}^{(K)}(\xi_a)$ completely characterize these transfer matrices. Let us introduce the functions

$$f_a^{(m)}(\lambda) = \prod_{b \neq a, b=1}^{N} \frac{\lambda - \xi_b}{\xi_a - \xi_b} \prod_{b=1}^{N} \prod_{r=1}^{m-1} \frac{1}{\xi_a - \xi_b^{(r)}}, \quad \xi_b^{(r)} = \xi_b - r\eta, \tag{2.78}$$

$$g_a^{(m)}(\lambda) = \prod_{b \neq a, b=1}^{N} \frac{\lambda - \xi_b}{\xi_a - \xi_b} \prod_{b=1}^{N} \prod_{r=1}^{m-1} \frac{1}{\xi_a - \xi_b^{(-r)}}, \tag{2.79}$$

and

$$T_{\infty,a}^{(K)}(\lambda) = T_{\infty,a}^{(K)} \prod_{b=1}^{N}(\lambda - \xi_b), \quad T_{\infty,(a)}^{(K)}(\lambda) = T_{\infty,(a)}^{(K)} \prod_{b=1}^{N}(\lambda - \xi_b), \tag{2.80}$$

then the following corollary holds:

**Corollary 2.1.** *Under the following conditions on the inhomogeneity parameters $\xi_i$*

$$\forall a, b \in \{1, \dots N\}, a \neq b, \quad \xi_a \neq \xi_b \mod \eta, \tag{2.81}$$

*the transfer matrix $T_{n+1}^{(K)}(\lambda)$ and $T_{(n+1)}^{(K)}(\lambda)$ are completely characterized in terms of $T_1^{(K)}(\lambda)$ by the fusion equations and the following interpolation formulae:*

$$T_{n+1}^{(K)}(\lambda) = \prod_{r=1}^{n} d(\lambda + r\eta) \left[ T_{\infty,n+1}^{(K)}(\lambda) + \sum_{a=1}^{N} f_a^{(n+1)}(\lambda) T_n^{(K)}(\xi_a + \eta) T_1^{(K)}(\xi_a) \right], \tag{2.82}$$

$$T_{(n+1)}^{(K)}(\lambda) = \prod_{r=1}^{n} d(\lambda - r\eta) \left[ T_{\infty,(n+1)}^{(K)}(\lambda) + \sum_{a=1}^{N} g_a^{(n+1)}(\lambda) T_{(n)}^{(K)}(\xi_a - \eta) T_1^{(K)}(\xi_a) \right]. \tag{2.83}$$

## 2.4 SoV covector basis for $gl_{\mathcal{M}|\mathcal{N}}$ Yang-Baxter superalgebra

In the next section, we construct a separation of variables basis for the integrable quantum model associated to the fundamental representations of the $gl_{\mathcal{M}|\mathcal{N}}$-graded Yang-Baxter algebra. The construction follows the general ideas presented in the Proposition 2.4 of [1].

As in the non-graded case, the proof relies mainly on the reduction of the $R$ matrix to the permutation at a particular point, and on the centrality of the asymptotics of the transfer matrix.

Let $K$ be a $(\mathcal{M} + \mathcal{N}) \times (\mathcal{M} + \mathcal{N})$ square matrix solution of the $gl_{\mathcal{M}|\mathcal{N}}$-graded Yang-Baxter equation of the block form (2.36), then we use the following notation

$$K = W_K K_J W_K^{-1}, \tag{2.84}$$

where $K_J$ is the Jordan form of the matrix $K$

$$K_J = \begin{pmatrix} K_{J,\mathcal{M}} & 0 \\ 0 & K_{J,\mathcal{N}} \end{pmatrix}, \tag{2.85}$$

where for $\mathcal{X} = \mathcal{M}$ or $\mathcal{N}$

$$K_{J,\mathcal{X}} = \begin{pmatrix} K_{J,\mathcal{X}}^{(1)} & 0 & \cdots & 0 \\ 0 & K_{J,\mathcal{X}}^{(2)} & \ddots & 0 \\ 0 & \ddots & \ddots & 0 \\ 0 & 0 & \cdots & K_{J,\mathcal{X}}^{(m_{\mathcal{X}})} \end{pmatrix}, \tag{2.86}$$

and $K_{J,\mathcal{X}}^{(i)}$ is a $d_{i,\mathcal{X}} \times d_{i,\mathcal{X}}$ upper triangular Jordan block for any $i$ in $\{1,...,m_{\mathcal{X}}\}$ with eigenvalue $k_{i,\mathcal{X}}$, where $\sum_{a=1}^{m_{\mathcal{X}}} d_{a,\mathcal{X}} = \mathcal{X}$. Moreover, it is interesting to point out that the invertible matrix $W_K$ defining the change of basis for the twist matrix is itself a $(\mathcal{M} + \mathcal{N}) \times (\mathcal{M} + \mathcal{N})$ square matrix solution of the $gl_{\mathcal{M}|\mathcal{N}}$-graded Yang-Baxter equation. Indeed, it is of the same block form (2.36):

$$W_K = \begin{pmatrix} W_{K,\mathcal{M}} & 0 \\ 0 & W_{K,\mathcal{N}} \end{pmatrix}. \tag{2.87}$$

Then, the following similarity relation holds for the fundamental transfer matrices:

$$T_1^{(K)}(\lambda) = \mathcal{W}_K T_1^{(K_J)}(\lambda)\mathcal{W}_K^{-1}, \quad \text{with } \mathcal{W}_K = W_{K,\mathsf{N}} \otimes \cdots \otimes W_{K,1}, \tag{2.88}$$

i.e. they are isospectral. We can now state our main result on the form of the SoV basis:

**Theorem 2.1.** *Let $K$ be a $(\mathcal{M} + \mathcal{N}) \times (\mathcal{M} + \mathcal{N})$ square matrix with simple spectrum of block form (2.36), i.e. we assume that:*

$$k_{i,\mathcal{X}} \neq k_{j,\mathcal{X}'} \text{ for } (i,\mathcal{X}) \neq (j,\mathcal{X}') \ \forall (i,j) \in \{1,...,m_{\mathcal{X}}\} \times \{1,...,m_{\mathcal{X}'}\}, \mathcal{X},\mathcal{X}' \in \{\mathcal{M},\mathcal{N}\}, \tag{2.89}$$

*then for almost any choice of the covector $\langle S|$ and of the inhomogeneities under the condition (2.81), the following set of covectors:*

$$\langle h_1,...,h_{\mathsf{N}}| \equiv \langle S| \prod_{n=1}^{\mathsf{N}} \left(T_1^{(K)}(\xi_n)\right)^{h_n} \text{ for any } \{h_1,...,h_{\mathsf{N}}\} \in \{0,...,\mathcal{M}+\mathcal{N}-1\}^{\times \mathsf{N}}, \tag{2.90}$$

*forms a covector basis of $\mathcal{H}$. In particular, let us take a one-site state $|S,a\rangle = S_i^{(a)}|i\rangle$, $S_i^{(a)} \in \mathbb{C}$. Its dual covector in the single space $V_a$ is $\langle S,a| = |S,a\rangle^\dagger = S_i^{(a)*}\langle i|$. When acting on it by the $W_K^{-1}$ isomorphism, it is noted in coordinates*

$$\langle S,a|W_{K,a}^{-1} = \left(S_1^{(a)*},...,S_{\mathcal{M}+\mathcal{N}}^{(a)*}\right) W_{K,a}^{-1} = \left(x_{1,\mathcal{M}}^{(1)},...,x_{d_1,\mathcal{M}}^{(1)},...,x_{1,\mathcal{N}}^{(m_{\mathcal{N}})},...,x_{d_{m_{\mathcal{N}}},\mathcal{N}}^{(m_{\mathcal{N}})}\right) \in V_a^*. \tag{2.91}$$

*Following (2.19), we have $\langle S| = (|S,1\rangle \ldots |S,\mathsf{N}\rangle)^\dagger$ as*

$$\langle S| = \sum_{p_1,...,p_{\mathsf{N}}=1}^{\mathcal{M}+\mathcal{N}} S_{p_1}^{(1)*} \ldots S_{p_{\mathsf{N}}}^{(\mathsf{N})*} (|p_1\rangle \ldots |p_{\mathsf{N}}\rangle)^\dagger. \tag{2.92}$$

*Then, it is sufficient that*

$$\prod_{k=1}^{m_{\mathcal{M}}} x_{1,\mathcal{M}}^{(k)} \prod_{k=1}^{m_{\mathcal{N}}} x_{1,\mathcal{N}}^{(k)} \neq 0, \tag{2.93}$$

*for the family of covectors (2.90) to form a basis. Furthermore, the $T_1^{(K)}(\lambda)$ transfer matrix spectrum is simple.*

*Proof.* As in the non graded case, the identity:

$$T_1^{(K)}(\xi_n) = R_{n,n-1}(\xi_n - \xi_{n-1}) \cdots R_{n,1}(\xi_n - \xi_1) K_n R_{n,N}(\xi_n - \xi_N) \cdots R_{n,n+1}(\xi_n - \xi_{n+1}), \quad (2.94)$$

holds true as a direct consequence of the definition of the transfer matrix $T_1^{(K)}(\lambda)$ and the properties

$$R_{0,n}(0) = \eta \mathbb{P}_{0,n}, \quad \mathrm{str}_{V_0} \mathbb{P}_{0,n} = 1. \quad (2.95)$$

From this point we can essentially follow the proof of Proposition 2.4 of [1]. Indeed, the condition that the set (2.90) form a covector basis of $\mathcal{H}$ is equivalent to

$$\det_{(\mathcal{M}+\mathcal{N})^N} ||\mathcal{R}(\langle S|, K, \{\xi\})|| \neq 0, \quad (2.96)$$

where we have defined:

$$\mathcal{R}(\langle S|, K, \{\xi\})_{i,j} \equiv \langle h_1(i), ..., h_N(i)|e_j\rangle, \quad \forall i, j \in \{1, ..., (\mathcal{M}+\mathcal{N})^N\}. \quad (2.97)$$

We are uniquely enumerating the $N$-tuple $(h_1(i), ..., h_N(i)) \in \{0, ..., \mathcal{M} + \mathcal{N} - 1\}^{\times N}$ by:

$$1 + \sum_{a=1}^{N} h_a(i)(\mathcal{M}+\mathcal{N})^{a-1} = i \in \{1, ..., (\mathcal{M}+\mathcal{N})^N\}, \quad (2.98)$$

and for any $j \in \{1, ..., (\mathcal{M}+\mathcal{N})^N\}$, $|e_j\rangle = |e_{1+h_1(j)}(1)\rangle \otimes ... \otimes |e_{1+h_N(j)}(N)\rangle \in \mathcal{H}$ is the corresponding element of the canonical basis in $\mathcal{H}$, where $|e_r(a)\rangle$ stands for the element $r \in \{1, ..., \mathcal{M} + \mathcal{N}\}$ of the canonical basis in the local quantum space $V_a$. Now, being the transfer matrix $T_1^{(K)}(\lambda)$ a polynomial in the inhomogeneities $\{\xi_i\}$ and in the parameters of the twist matrix $K$, the same statement holds true for the determinant on the l.h.s. of (2.96), which is moreover a polynomial in the coefficients $\langle S|e_j\rangle$ of the covector $\langle S|$.

Then it follows that the condition (2.96) holds true for almost any value of these parameters if one can show it under the special limit of large inhomogeneities. Using this argument, the form of the transfer matrix in the inhomogeneities (2.94) and the central asymptotics of the $gl_{\mathcal{M}|\mathcal{N}}$-graded $R$-matrix one can show that a sufficient criterion is that the following determinant is non-zero:

$$\det_{(\mathcal{M}+\mathcal{N})^N} ||\left(\langle S|K_1^{h_1(i)} \cdots K_N^{h_N(i)}|e_j\rangle\right)_{i,j \in \{1, ..., (\mathcal{M}+\mathcal{N})^N\}}|| \neq 0. \quad (2.99)$$

Let us compute this matrix element precisely: from (2.92), it decomposes as the following sum

$$\langle S|K_1^{h_1(i)} \ldots K_N^{h_N(i)}|e_j\rangle = \quad (2.100)$$
$$\sum_{p_1, ..., p_N=1}^{\mathcal{M}+\mathcal{N}} S_{p_1}^{(1)*} \ldots S_{p_N}^{(N)*} |p_1 \ldots p_N\rangle^\dagger K_1^{h_1(i)} \ldots K_N^{h_N(i)} |e_{1+h_1(j)}(1)\rangle \ldots |e_{1+h_N(j)}(N)\rangle.$$

Now, the $K_a^{h_a(i)}$ being even, the matrix element factorizes by (2.21) as a product over the one site matrix elements

$$|p_1 \ldots p_N\rangle^\dagger K_1^{h_1(i)} \ldots K_N^{h_N(i)} |e_{1+h_1(j)}(1)\rangle \ldots |e_{1+h_N(j)}(N)\rangle$$
$$= \langle p_1|K_1^{h_1(i)}|e_{1+h_1(j)}(1)\rangle \ldots \langle p_N|K_N^{h_N(i)}|e_{1+h_N(j)}(N)\rangle. \quad (2.101)$$

Therefore the sum over $p_1, ..., p_N$ decouples as a product of N sums, and identifying $\langle S, a| = S_{p_a}^{(a)*} \langle p_a|$ in the expression leaves us with

$$\langle S|K_1^{h_1(i)} \ldots K_N^{h_N(i)}|e_j\rangle = \langle S, 1|K^{h_1(i)}|e_{1+h_1(j)}(1)\rangle \ldots \langle S, N|K^{h_N(i)}|e_{1+h_N(j)}(N)\rangle. \quad (2.102)$$

Hence, the determinant factorizes and the criterion amounts to

$$\prod_{a=1}^{N} \det_{\mathcal{M}+\mathcal{N}} ||\left( \langle S, a | K_a^{i-1} | e_j(a) \rangle \right)_{i,j \in \{1,...,\mathcal{M}+\mathcal{N}\}} || \neq 0. \tag{2.103}$$

Finally, by Proposition 2.2 of [1], it holds for the factor corresponding to site $n$ in the above product

$$\det_{\mathcal{M}+\mathcal{N}} || \langle S, n | K_n^{i-1} | e_j(n) \rangle_{i,j \in \{1,...,\mathcal{M}+\mathcal{N}\}} || =$$
$$\prod_{a=1}^{m_{\mathcal{M}}} \left( x_{1,\mathcal{M}}^{(a)} \right)^{d_{a,\mathcal{M}}} \prod_{a=1}^{m_{\mathcal{N}}} \left( x_{1,\mathcal{N}}^{(a)} \right)^{d_{a,\mathcal{N}}} \prod_{a=1}^{m_{\mathcal{M}}} \prod_{b=1}^{m_{\mathcal{N}}} \left( k_{a,\mathcal{M}} - k_{b,\mathcal{N}} \right)^{d_{a,\mathcal{M}} d_{b,\mathcal{N}}}$$
$$\times \prod_{1 \leq a < b \leq m_{\mathcal{M}}} \left( k_{a,\mathcal{M}} - k_{b,\mathcal{M}} \right)^{d_{a,\mathcal{M}} d_{b,\mathcal{M}}} \prod_{1 \leq a < b \leq m_{\mathcal{N}}} \left( k_{a,\mathcal{N}} - k_{b,\mathcal{N}} \right)^{d_{a,\mathcal{N}} d_{b,\mathcal{N}}}, \tag{2.104}$$

which is clearly nonzero under the condition that the twist $K$ has simple spectrum and that (2.93) holds. The simplicity of the transfer matrix spectrum is then a trivial consequence of the fact that the set of covectors (2.90) is proven to be a basis. Indeed, it implies that given a generic eigenvalue $t(\lambda)$ of $T_1^{(K)}(\lambda)$, the associated eigenvector $|t\rangle$ is unique, being characterized uniquely (up to normalization) by the eigenvalue as

$$\langle h_1, ..., h_N | t \rangle = \prod_{a=1}^{N} t^{h_a}(\xi_a), \quad \forall (h_1, ..., h_N) \in \{0, ..., \mathcal{M} + \mathcal{N} - 1\}^{\times N}. \tag{2.105}$$

$\square$

*Remark* 2.2. Note that $\langle S | \neq \langle S, 1 | \ldots \langle S, N |$.

The norm of $|S\rangle$ is $\langle S | S \rangle = \prod_{a=1}^{N} \sum_{i=1}^{\mathcal{M}+\mathcal{N}} |S_i|^2$ and can be set to convenience. In particular, it may be taken to one.

Let us observe that some stronger statement can be done about the transfer matrix diagonalizability and spectrum simplicity according to the following

**Proposition 2.1.** *Let the twist matrix $K$ be diagonalizable and with simple spectrum on $\mathbb{C}^{\mathcal{M}|\mathcal{N}}$, then $T_1^{(K)}(\lambda)$ is diagonalizable and with simple spectrum, for almost any values of the inhomogeneities satisfying the condition (2.81). Indeed, taken the generic eigenvalue $t(\lambda)$ of $T_1^{(K)}(\lambda)$ it holds:*

$$\langle t | t \rangle \neq 0, \tag{2.106}$$

*where $|t\rangle$ and $\langle t|$ are the unique eigenvector and eigencovector associated to it.*

*Proof.* Following the proof of Proposition 2.5 of [1] the non-orthogonality condition (2.106) can be derived. Such condition together with the simplicity of the spectrum implies that we cannot have non-trivial Jordan blocks in the transfer matrix spectrum so that it must be diagonalizable and with simple spectrum. $\square$

## 2.5 On closure relations and SoV spectrum characterization

In the previous two subsections, we have shown how the transfer matrix $T_1^{(K)}(\lambda)$ associated to general inhomogeneous representations of the $gl_{\mathcal{M}|\mathcal{N}}$-graded Yang-Baxter algebra allows to reconstruct all the fused transfer matrices (mainly by using the known fusion relations (2.65) and (2.66)). Moreover, we have shown that $T_1^{(K)}(\lambda)$ allows to characterize an SoV basis, which also implies its spectrum simplicity or diagonalizability and spectrum simplicity if the twist matrix $K$ is, respectively, with simple spectrum or diagonalizable with simple spectrum.

This analysis shows that the full integrable structure of the $gl_{\mathcal{M}|\mathcal{N}}$-graded Yang-Baxter algebra can be recast in its fundamental transfer matrix as well as the construction of quantum separation of variables. However, there is still one missing information, which is a functional equation, or a discrete system of equations, allowing the complete characterization of the transfer matrix spectrum. As already mentioned, the fusion relations (2.51) alone only give the characterization of higher transfer matrices in terms of the first one. Some further algebra and representation dependent rules are required in order to complete them and extract a closure relation on the transfer matrix.

In the case of the quantum integrable models associated to the fundamental representations of the $gl_{\mathcal{M}}$ and $U_q(\widehat{gl_{\mathcal{M}}})$ Yang-Baxter and reflection algebras, such a closure relation comes from the quantum determinant [1,45,46,159–161]. Indeed, $P^-_{1\cdots\mathcal{M}}$ is a rank 1 projector in these cases, implying that the corresponding transfer matrix $T^{(K)}_{(\mathcal{M})}(\lambda)$ becomes a computable central element of the Yang-Baxter algebra, namely the quantum determinant. Then, substituting the quantum determinant in the fusion equation (2.65) for $n = \mathcal{M} - 1$ and using the same interpolation formulae for the higher fused transfer matrix eigenvalues, we produce a discrete system of polynomial equations with N equations in N unknowns which was proven [1,45,46,159–161] to completely characterize the transfer matrix spectrum in quantum separation of variables. In the case of non-fundamental representations[13] of the same algebras the closure relation comes instead with the appearing of the first central zeros in the fused transfer matrices of type $T^{(K)}_n(\lambda)$. In [162], this analysis has been developed in detail in the case of $\mathcal{M} = 2$. There, it has been shown that imposing the central zeros of the fused transfer matrix $T^{(K)}_{2s+1}(\lambda)$, for a spin $s \geq 1$ representation, a discrete system of polynomial equations with N equations in N unknowns is derived for the transfer matrix eigenvalues. The set of its solutions completely characterizes the transfer matrix spectrum in quantum separation of variables. In the nonfundamental and cyclic representations of the $U_q(\widehat{gl_{\mathcal{M}}})$ Yang-Baxter algebra for $q$ a root of unity such closure relation comes from the so-called truncation identities. For $\mathcal{M} = 2$, it has been shown in [40] how these identities emerge and are proven in the framework of the quantum separation of variables and how they are used to completely characterize the transfer matrix spectrum. In [44] and [59,60] these results have been extended, respectively, to the most general cyclic representations of the $U_q(\widehat{gl_2})$ Yang-Baxter algebra and reflection algebra.

In the case of integrable quantum lattice models associated to the fundamental representations of the $gl_{\mathcal{M}|\mathcal{N}}$-graded Yang-Baxter algebra, the natural candidate for the closure relation is the *inner-boundary* condition (2.54). Indeed, once we impose it on the eigenvalues $t^{(\mathcal{M}+1),(K)}_{\mathcal{N}}(\lambda)$ and $t^{(\mathcal{M}),(K)}_{\mathcal{N}+1}(\lambda)$ of the transfer matrices $T^{(\mathcal{M}+1),(K)}_{\mathcal{N}}(\lambda)$ and $T^{(\mathcal{M}),(K)}_{\mathcal{N}+1}(\lambda)$, we are left with one nontrivial functional equation containing as unknowns the eigenvalues of the first transfer matrix computed in the inhomogeneities $t^{(K)}_1(\xi_{i\leq \mathsf{N}})$. This is the case as the eigenvalues $t^{(\mathcal{M}+1),(K)}_{\mathcal{N}}(\lambda)$ and $t^{(\mathcal{M}),(K)}_{\mathcal{N}+1}(\lambda)$ admit the same expansion in terms of the transfer matrix eigenvalue $t^{(K)}_1(\lambda)$ as those derived in subsection 2.3 for the transfer matrices $T^{(\mathcal{M}+1),(K)}_{\mathcal{N}}(\lambda)$ and $T^{(\mathcal{M}),(K)}_{\mathcal{N}+1}(\lambda)$ in terms of the transfer matrix $T^{(K)}_1(\lambda)$. Moreover, the *inner-boundary* condition (2.54) involved the quantum Berezinian as a central element hence playing a role similar to the quantum determinant in the bosonic case. More precisely, we can introduce the following polynomials:

$$t_1(\lambda|\{x_a\}) = T^{(K)}_{\infty,1}(\lambda) + \sum_{a=1}^{\mathsf{N}} f^{(1)}_a(\lambda) x_a, \qquad (2.107)$$

---

[13]Here $q$ is not a root of unity for the quantum group case.

and from them recursively the following higher polynomials

$$t_{n+1}(\lambda|\{x_a\}) = \prod_{r=1}^{n} d(\lambda + r\eta)\left[T_{\infty,n+1}^{(K)}(\lambda) + \sum_{a=1}^{N} f_a^{(n+1)}(\lambda)t_n(\xi_a + \eta|\{x_a\})x_a\right], \qquad (2.108)$$

$$t_{(n+1)}(\lambda|\{x_a\}) = \prod_{r=1}^{n} d(\lambda - r\eta)\left[T_{\infty,(n+1)}^{(K)}(\lambda) + \sum_{a=1}^{N} g_a^{(n+1)}(\lambda)t_{(n)}(\xi_a - \eta|\{x_a\})x_a\right], \quad (2.109)$$

and

$$t_b^{(a)}(\lambda|\{x_a\}) = \det_{1\leq i,j\leq a} t_{b+i-j}(\lambda - (i-1)\eta|\{x_a\}) \qquad (2.110)$$

$$= \det_{1\leq i,j\leq b} t_{(a+i-j)}(\lambda + (i-1)\eta|\{x_a\}). \qquad (2.111)$$

Then the following lemma holds

**Lemma 2.2.** *Any transfer matrix $T_1^{(K)}(\lambda)$ eigenvalue[14] admits the representation $t_1(\lambda|\{x_a\})$, where the $\{x_a\}$ are solutions of the inner-boundary condition (2.54):*

$$(-1)^{\mathcal{N}} Ber(\lambda)t_{\mathcal{N}}^{(\mathcal{M}+1)}(\lambda + \eta|\{x_a\}) = t_{\mathcal{N}+1}^{(\mathcal{M})}(\lambda|\{x_a\}), \quad \forall \lambda \in \mathbb{C}, \qquad (2.112)$$

*and of the null out-boundary conditions (2.53):*

$$t_{\mathcal{N}+m}^{(\mathcal{M}+n)}(\lambda|\{x_a\}) = 0, \quad \forall \lambda \in \mathbb{C} \text{ and } n, m \geq 1. \qquad (2.113)$$

In the next section, we conjecture that the above system of functional equations completely characterizes the transfer matrix spectrum in the case of the $gl_{1|2}$-graded Yang-Baxter algebra. We prove this characterization for some special class of twist matrices while we only give some first motivations of it for general representations. In appendix B we verify it for quantum chains with two and three sites. Let us also mention that this conjecture can be checked explicitly for the simple $gl_{1|1}$ case. It would be interesting in this respect to elucidate the relation of our method with the one developed recently in [204].

# 3   On SoV spectrum description of $gl_{1|2}$ Yang-Baxter superalgebra

Specialising to the $gl_{1|2}$ case, some results have already been obtained in the context of the NABA, see [136, 205] for instance.

## 3.1   General statements and conjectured closure relation for general integrable twist

We use this subsection to clarify and justify the following conjecture

**Conjecture 3.1.** *Taken the general $gl_{1|2}$-graded Yang-Baxter algebra with twisted boundary conditions, the polynomial $t_1(\lambda|\{x_a\})$ defined above is an eigenvalue of the transfer matrix $T_1^{(K)}(\lambda)$ (excluding the trivial solution $x_1 = \ldots = x_N = 0$) iff: the higher polynomials associated to it satisfy, in addition to the fusion relations, the inner-boundary condition (2.112) and the null out-boundary conditions (2.113) for $\mathcal{M} = 1$ and $\mathcal{N} = 2$.*

The fat hook domain for $gl_{1|2}$ is pictured in figure 3. In the $gl_{1|2}$-graded case under con-

---

[14]The trivial solution $\{x_1, ..., x_N\} = \{0, ..., 0\}$ has to be excluded for invertible twist matrix.

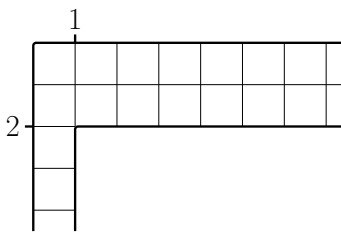

Figure 3: Admissible domain for Young Diagrams of $gl_{1|2}$.

sideration the inner-boundary condition (2.54) reads:

$$T_2^{(2),(K)}(\lambda+\eta)k_1 = k_2 k_3 d(\lambda) T_3^{(K)}(\lambda), \tag{3.1}$$

where:

$$\det K_{\mathcal{M}=1} = K_{\mathcal{M}=1} = k_1 \text{ and } \det K_{\mathcal{N}=2} = k_2 k_3. \tag{3.2}$$

Moreover, we have the following expressions[15] for the asymptotics of the fused transfer matrices

$$T_{\infty,n}^{(K)} = k_1^{n-2}(k_1-k_3)(k_1-k_2), \quad \forall n \geq 2. \tag{3.3}$$

Then imposing it for the corresponding eigenvalues we get

$$k_3 k_2 d(\lambda) t_3(\lambda) = k_1 t_2^{(2)}(\lambda+\eta), \tag{3.4}$$

which, once we express the eigenvalues $t_2^{(2)}(\lambda+\eta)$ by the use of the fusion relation (2.51):

$$t_2^{(2)}(\lambda+\eta) = t_2(\lambda)t_2(\lambda+\eta) - t_1(\lambda+\eta)t_3(\lambda), \tag{3.5}$$

takes the following closure relation form:

$$k_3 k_2 d(\lambda) t_3(\lambda) = k_1(t_2(\lambda)t_2(\lambda+\eta) - t_3(\lambda)t_1(\lambda+\eta)). \tag{3.6}$$

Now, using the interpolation formulae (2.107) and (2.108) for the transfer matrix eigenvalues, we get that the closure relation is indeed a functional equation whose unknowns coincide with the $x_{i\leq N} \equiv t_1^{(K)}(\xi_{i\leq N})$. The transfer matrix eigenvalues have to satisfy furthermore the null out-boundary conditions (2.53), which reads:

$$t_{3+m}^{(2+n)}(\lambda) = 0, \quad \forall \lambda \in \mathbb{C} \text{ and } n, m \geq 0, \tag{3.7}$$

in the $gl_{1|2}$-graded case under consideration.

According to our Conjecture the transfer matrix spectrum coincides with the set of solutions to the functional equations (3.6) and (3.7) in the unknowns $t_1^{(K)}(\xi_{i\leq N})$. This spectrum characterization for general twist matrix will be proven by direct action of the transfer matrices on our SoV basis in our next publication. Here we present some arguments in favour of it while in the next subsections 3.2 we prove it for a special choice of the twist matrix.

Let us consider the case of a twist matrix $K$ invertible with simple spectrum[16], then our SoV basis can be written as follows:

$$\langle h_1,...,h_N| \equiv \langle \bar{s}| \prod_{n=1}^{N}(T_1^{(K)}(\xi_n))^{h_n} \text{ for any } \{h_1,...,h_N\} \in \{1,2,3\}^{\times N}, \tag{3.8}$$

---

[15]They can be computed for example by induction starting from the explicit formulae for $T_{\infty,1}^{(K)}$ and $T_{\infty,2}^{(K)}$, by using the fusion equations and the null out-boundary conditions.

[16]Indeed, the case of $K$ non-invertible but having simple spectrum of the form (3.12) will be described in detail in the next subsection.

indeed in this case the transfer matrices $T_1^{(K)}(\xi_n)$ are invertible[17] and so we can write our original vector $\langle S|$ as it follows:

$$\langle S| = \langle \bar{S}| \prod_{n=1}^{\mathsf{N}} T_1^{(K)}(\xi_n). \tag{3.9}$$

If the $t_1(\xi_{i\leq\mathsf{N}})$ solve the closure relation (3.6) and the null out-boundary conditions (3.7), to prove that a vector $|t\rangle$ characterized by

$$\langle h_1, ..., h_{\mathsf{N}}|t\rangle \equiv \prod_{n=1}^{\mathsf{N}} t_1^{h_n}(\xi_n) \quad \forall \{h_1, ..., h_{\mathsf{N}}\} \in \{1, 2, 3\}^{\times \mathsf{N}}, \tag{3.10}$$

is indeed a transfer matrix eigenvector, the main point is to be able to reduce the covectors containing a fourth order power of $T_1^{(K)}(\xi_n)$ in those of the SoV basis, of maximal order three. Moreover, this reduction must come from relations which are satisfied identically by both the fused transfer matrices and the functions $t_r(\lambda|\{x_a\})$ defined in (2.108), in terms of the given solution $t_1(\xi_{i\leq\mathsf{N}})$. Indeed, this is exactly what it is done by the closure relation for the transfer matrix:

$$k_3 k_2 d(\lambda) T_3^{(K)}(\lambda) = k_1 (T_2^{(K)}(\lambda) T_2^{(K)}(\lambda + \eta) - T_3^{(K)}(\lambda) T_1^{(K)}(\lambda + \eta)), \tag{3.11}$$

and by the corresponding one (3.6) for the eigenvalues. As one can easily remark that (3.11) and (3.6) are both of fourth order, respectively, in the $T_1^{(K)}(\xi_n)$ and $t_1(\xi_n)$ on the right hand side while they are both of third order on the left hand side.

In appendix A, we will verify that Nested Algebraic and Analytic Bethe Ansatz are indeed compatible with these requirements, i.e. the functional ansatz for the eigenvalues $t_1(\lambda)$ indeed satisfies the closure relation (3.6) and the null out-boundary conditions (3.7). There, we moreover argue the completeness of the Bethe Ansatz which is compatible with our Conjecture.

It is also worth to mention that we have verified our Conjecture on small lattices, up to three sites. More in detail, we have solved the discrete system of $\mathsf{N}$ equations in the $\mathsf{N}$ unknowns $t_1^{(K)}(\xi_{i\leq\mathsf{N}})$ obtained particularizing (3.6) in $\mathsf{N}$ distinct values of $\lambda$. Among these solutions we have selected the solutions verifying the null out-boundary conditions (3.7) for $n = m = 0$. This has produced exactly $3^{\mathsf{N}}$ distinct solutions which are proven to coincide with $T_1^{(K)}(\lambda)$ transfer matrix eigenvalues computed by direct diagonalization, see appendix B.

## 3.2 SoV spectrum characterization for non-invertible and simple spectrum twist matrix

Let us study here the spectral problem for the transfer matrices associated to the fundamental representations of the $gl_{1|2}$-graded Yang-Baxter algebra in the following class of non-invertible but having simple spectrum $\hat{K}$ twist matrices:

$$\hat{K} = \begin{pmatrix} k_1 = 0 & 0 \\ 0 & K_2 \end{pmatrix}_{3\times3}, \tag{3.12}$$

with $K_2$ any invertible, diagonalizable and simple $2 \times 2$ matrix, i.e. it holds:

$$k_2 \neq k_3, \ k_i \neq 0, \ i = 2, 3. \tag{3.13}$$

Despite $K$ having a zero eigenvalue, the results of subsection 2.4 imply that the set of covectors (2.90) still forms a covector basis of $\mathscr{H}$. Moreover, these are non-trivial fundamental

---

[17]The reconstruction of local operators, pioneered in [87,88], implies that the twisted transfer matrix computed in the inhomogeneities coincides with the local matrix K at the site n dressed by the product of shift operators along the chain. So, they are invertible as all these operators are invertible.

representations of the $gl_{1|2}$-graded Yang-Baxter algebra for which our Conjecture is verified, as shown in the following:

**Theorem 3.1.** *For almost any values of the inhomogeneities $\{\xi_{a\leq N}\}$ satisfying the condition* (2.81) *and the twist matrix eigenvalues satisfying* (3.13)*, the eigenvalue spectrum of $T_1^{(\hat{K})}(\lambda)$ coincides with the following set of polynomials*

$$\Sigma_{T^{(\hat{k})}} = \left\{ t_1(\lambda) : t_1(\lambda) = -(k_2 + k_3) \prod_{a=1}^{N}(\lambda - \xi_a) + \sum_{a=1}^{N} f_a^{(1)}(\lambda)x_a, \quad \forall \{x_1,...,x_N\} \in S_{T^{(\hat{k})}} \right\}, \tag{3.14}$$

*where $S_{T^{(\hat{k})}}$ is the set of solutions to the following system of N cubic equations:*

$$x_a \left[ k_2 k_3 d(\xi_a + \eta) + \sum_{r=1}^{N} f_r^{(2)}(\xi_a + \eta)t_1(\xi_r + \eta)x_r \right] = 0, \quad \forall a \in \{1,...,N\}, \tag{3.15}$$

*in N unknown $\{x_1,...,x_N\}$. Moreover, $T_1^{(\hat{K})}(\lambda)$ is diagonalizable and with simple spectrum. For any $t_1(\lambda) \in \Sigma_{T^{(\hat{k})}}$, the associated and unique eigenvector $|t\rangle$ (up-to normalization) has the following wave-functions in our SoV covector basis:*

$$\langle h_1,...,h_N|t\rangle = \prod_{n=1}^{N} t_1^{h_n}(\xi_n). \tag{3.16}$$

*Proof.* The main identity to be pointed out here is the following one:

$$T_3^{(\hat{K})}(\lambda) \equiv 0, \tag{3.17}$$

due to the closure relation (3.11) being $k_1 = 0$. So that the fusion equations (2.66) for $n = 1$ and $n = 2$ read:

$$T_2^{(\hat{K})}(\xi_a) = T_1^{(\hat{K})}(\xi_a)T_1^{(\hat{K})}(\xi_a + \eta), \tag{3.18}$$

$$0 = T_1^{(\hat{K})}(\xi_a)T_2^{(\hat{K})}(\xi_a + \eta). \tag{3.19}$$

Now it is easy to verify that the system of equations (3.15) just coincides with the above fusion conditions once imposed to functions which have the analytic properties (polynomial form and asymptotics) of eigenvalues. So that it is clear that any eigenvalue has to satisfy them and one is left with the proof of the reverse statement. This proof can be done just showing that the state $|t\rangle$ of the form (3.16) is indeed an eigenvector of the transfer matrix, i.e. that it holds:

$$\langle h_1,...,h_N|T_1^{(\hat{K})}(\lambda)|t\rangle = t_1(\lambda)\langle h_1,...,h_N|t\rangle, \tag{3.20}$$

by direct action of the transfer matrix $T_1^{(\hat{K})}(\lambda)$ on the SoV basis. The steps of the proof are indeed completely similar to those described in the proof of Theorem 5.1 of [1].

Finally, let us point out that Proposition 2.1 implies also that the transfer matrix $T_1^{(\hat{K})}(\lambda)$ is diagonalizable and with simple spectrum for general values of the inhomogeneities parameters. $\square$

*Remark* 3.1. It is important to point out that the above theorem proves the validity of our Conjecture for the representations considered here, as the system of equations (3.15) is equivalent to the conjectured characterization given by the functional equations (3.6) and (3.7) in the unknowns $t_1^{(K)}(\xi_{i\leq N})$. Indeed, the system of equations (3.15) is just equivalent to the functional equation

$$t_3(\lambda|\{t_1^{(K)}(\xi_{i\leq N})\}) = 0, \tag{3.21}$$

which coincides with the closure relation (3.6) for the $k_1 = 0$ case under consideration. Then it is easy to verify that the null out-boundary conditions (3.7) are also verified. Note for example that the condition (3.21) together with the interpolation formula (2.108) for $k_1 = 0$ implies:

$$t_n(\lambda|\{t_1^{(K)}(\xi_{i\leq N})\}) = 0, \quad \forall n \geq 3, \tag{3.22}$$

so that the interpolation formula (2.110) implies:

$$t_3^{(2)}(\lambda|\{t_1^{(K)}(\xi_{i\leq N})\}) = t_3(\lambda - \eta|\{t_1^{(K)}(\xi_{i\leq N})\})t_3(\lambda|\{t_1^{(K)}(\xi_{i\leq N})\}) - t_4(\lambda - \eta|\{t_1^{(K)}(\xi_{i\leq N})\})$$
$$\times t_2(\lambda|\{t_1^{(K)}(\xi_{i\leq N})\}) \tag{3.23}$$
$$= 0, \tag{3.24}$$

i.e. the null out-boundary conditions (3.7) for $n = m = 0$ and similarly for the others.

*Remark* 3.2. Let us comment that a different proof of the above theorem can be given by using the fact that the transfer matrix $T_1^{(\hat{K})}(\lambda)$ is diagonalizable with simple spectrum. This in particular means that this transfer matrix admits $3^N$ distinct eigenvalues anyone being a solution of the system (3.15) of N polynomial equations of order three in N unknowns. The Theorem of Bézout[18] states that the above system of polynomial equations admits $3^N$ solutions if the N polynomials, defining the system, have no common components.[19] So, under the condition of no common components, there are indeed exactly $3^N$ distinct solutions to the above system and each one is uniquely associated to a transfer matrix eigenvalue. The proof of the condition of no common components can be done following exactly the same steps presented in appendix B of [160].

*Remark* 3.3. The fact that $T_3^{(\hat{K})}(\lambda)$ is identically zero in these representations associated to non-invertible simple spectrum twist matrix $\hat{K}$ means, in particular, that it is central so that the algebra shows some strong resemblance to the twisted representations of the $gl_3$ Yang-Baxter algebra. In fact, taking the $gl_3$-representation associated to the twist matrix $K' = -\hat{K}$ and $\eta' = -\eta$, then the SoV characterization of the spectrum implies that the transfer matrix $T_1^{(\hat{K})}(\lambda|\eta)$, associated to the $gl_{1|2}$-representation, is isospectral to the transfer matrix $T_1^{(-\hat{K})}(\lambda|-\eta)$, associated to the $gl_3$-representation. In the same way, we have that $T_2^{(\hat{K})}(\lambda|\eta)$, associated to the $gl_{1|2}$-representation, is isospectral to $T_{(2)}^{(-\hat{K})}(\lambda|-\eta)$, associated to the $gl_3$-representation.

It is worth remarking that the same type of duality indeed holds between the $gl_{1|\mathcal{N}}$-graded and the $gl_{\mathcal{N}+1}$ non-graded Yang-Baxter algebra when associated to the non-invertible but simple $(\mathcal{N}+1) \times (\mathcal{N}+1)$ twist matrix $\hat{K}$ with first eigenvalue zero. More in detail, we have the isospectrality of the transfer matrices $T_m^{(\hat{K})}(\lambda|\eta)$, associated to the $gl_{1|\mathcal{N}}$-representation, with the transfer matrices $T_{(m)}^{(-\hat{K})}(\lambda|-\eta)$, associated to the $gl_{\mathcal{N}+1}$-representation, for any $1 \leq m \leq \mathcal{N}$. This in particular implies that we can characterize completely as well the spectrum of the transfer matrices of the $gl_{1|\mathcal{N}}$-graded Yang-Baxter algebra for this special class of twist matrices just using the results of [159]. Then the results of the next two subsections can be as well generalized to these special classes of $gl_{1|\mathcal{N}}$-graded Yang-Baxter algebras.

### 3.2.1 The quantum spectral curve equation for non-invertible twist

The transfer matrix spectrum in our SoV basis is equivalent to the quantum spectral curve[20] functional reformulation as stated in the next theorem.

---

[18]See for example William Fulton (1974). Algebraic Curves. Mathematics Lecture Note Series. W.A. Benjamin.

[19]Indeed, if there are common components the system admits instead an infinite number of solutions.

[20]To our knowledge, the quantum spectral curve terminology has been introduced by Sklyanin, see for example [25]. It comes natural as the transfer matrices can be seen as the quantum counterpart of the spectral invariants of

**Theorem 3.2.** *Under the same conditions of the previous theorem, then an entire function $t_1(\lambda)$ is a $T_1^{(\hat{K})}(\lambda)$ transfer matrix eigenvalue iff. there exists a unique polynomial:*

$$\varphi_t(\lambda) = \prod_{a=1}^{M} (\lambda - \nu_a) \ \text{with} \ M \leq N \ \text{and} \ \nu_a \neq \xi_n \ \forall (a, n) \in \{1, ..., M\} \times \{1, ..., N\}, \qquad (3.25)$$

*such that $t_1(\lambda)$, $t_2(\lambda|\{t_1(\xi_{a\leq N})\})$ and $\varphi_t(\lambda)$ are solutions of the following quantum spectral curve functional equation:*

$$\varphi_t(\lambda - \eta)t_2(\lambda - \eta) + \alpha(\lambda)\varphi_t(\lambda)t_1(\lambda) + \beta(\lambda)\varphi_t(\lambda + \eta) = 0, \qquad (3.26)$$

*where we have defined*

$$\alpha(\lambda) = -\bar{\alpha} \prod_{a=1}^{N} (\lambda - 2\eta - \xi_a), \quad \beta(\lambda) = \alpha(\lambda)\alpha(\lambda + \eta), \qquad (3.27)$$

*and $\bar{\alpha}$ is a nonzero solution of the characteristic equation:*

$$\bar{\alpha} T_{\infty,2}^{(\hat{K})} - \bar{\alpha}^2 T_{\infty,1}^{(\hat{K})} + \bar{\alpha}^3 \mathbb{I}_{\mathscr{H}} = 0, \qquad (3.28)$$

*i.e. $\bar{\alpha} = -k_2$ or $\bar{\alpha} = -k_3$ is a nonzero eigenvalue of the twist matrix $-\hat{K}$. Moreover, up to a normalization, the common transfer matrix eigenvector $|t\rangle$ admits the following separate representation:*[21]

$$\langle h_1, ..., h_N | t \rangle = \prod_{a=1}^{N} \alpha^{h_a}(\xi_a + \eta)\varphi_t^{h_a}(\xi_a + \eta)\varphi_t^{2-h_a}(\xi_a). \qquad (3.29)$$

*Proof.* From the above Remark 3.3, we have that this theorem is a direct consequence of the Theorem 5.2 of [1]. To make the comparison easier, one has just to take the quantum spectral curve characterization of Theorem 4.1 of [159] and use it in the case $n = 3$, $K \to -\hat{K}$, $\eta \to -\eta$ to get the quantum spectral curve associated to the non-invertible simple spectrum twist $\hat{K}$. Here, $t_1(\lambda)$ is the eigenvalue associated to the $gl_{1|2}$-transfer matrix $T_1^{(\hat{K})}(\lambda|\eta)$, isospectral to the $gl_3$-transfer matrix $T_1^{(-\hat{K})}(\lambda| - \eta)$, and $t_2(\lambda|\{t_1(\xi_{a\leq N})\})$ is the eigenvalue of the $gl_{1|2}$-transfer matrix $T_2^{(\hat{K})}(\lambda|\eta)$, isospectral to the $gl_3$-transfer matrix $T_{(2)}^{(-\hat{K})}(\lambda| - \eta)$. Then, just removing the common zeros, in the three nonzero terms of the equation and making the common shift $\lambda \to \lambda - 2\eta$, we get our quantum spectral curve equation (3.26). Let us recall that the main elements in the proof of the theorem rely on the fact that the quantum spectral curve

---

the monodromy matrix. In fact, in [25], these are operatorial functional equations involving just one $Q$-operator, the canonical operators (i.e. the separate variable operators) and the exponential of their canonical conjugated operators (i.e. the shift operators) and the quantum spectral invariants of the monodromy matrix. In general, we write the quantum spectral curve in its coordinate form, i.e. our quantum spectral curve can be seen as the matrix element of the Sklyanin's one between a transfer matrix eigenstate and an SoV basis element, when Sklyanin's SoV applies, otherwise our results generalize Sklyanin's ones. However, in general, the fact that we can prove that the transfer matrix has simple spectrum and it is diagonalizable allows us to rewrite these quantum spectral curves at the operator level.

[21]Note that (3.29) can be seen as a rewriting of the transfer matrix eigen-wavefunctions in terms of the eigenvalues of a $Q$-operator. In fact, for $k_1 = 0$, the equation (3.54) and (3.58) imply that the functions $\varphi_t(\lambda)$ are strictly related to the eigenvalues of the operator $Q_2(\lambda)$. Similarly for $k_1 \neq 0$, by using the NABA expression (3.45) and the original SoV representation of the transfer matrix eigen-wavefunctions (3.16), one can argue that (3.29) should be true with $\bar{\alpha} = k_1$ and $\varphi_t(\lambda)$ coinciding with the eigenvalues of the operator $Q_1(\lambda)$.

is equivalent to the following 3N conditions:

$$\alpha(\xi_a + \eta)\frac{\varphi_t(\xi_a + \eta)}{\varphi_t(\xi_a)} = t_1(\xi_a), \quad \text{for } \lambda = \xi_a, \ \forall a \in \{1, ..., N\}, \tag{3.30}$$

$$\alpha(\xi_a + \eta)\frac{\varphi_t(\xi_a + \eta)}{\varphi_t(\xi_a)} = \frac{t_2(\xi_a)}{t_1(\xi_a + \eta)}, \quad \text{for } \lambda = \xi_a + \eta, \ \forall a \in \{1, ..., N\}, \tag{3.31}$$

$$\varphi_t(\xi_a + \eta)t_2(\xi_a + \eta) = 0, \quad \text{for } \lambda = \xi_a + 2\eta, \ \forall a \in \{1, ..., N\}, \tag{3.32}$$

once the asymptotics are fixed as stated above in this theorem. It is then easy to observe that the compatibility of this system of equations is equivalent to impose that $t_1(\lambda)$ and $t_2(\lambda|\{t_1(\xi_{a \leq N})\})$ satisfy the fusion equations:

$$t_1(\xi_a)t_1(\xi_a + \eta) = t_2(\xi_a), \quad \forall a \in \{1, ..., N\}, \tag{3.33}$$

$$t_1(\xi_a)t_2(\xi_a + \eta) = 0, \quad \forall a \in \{1, ..., N\}. \tag{3.34}$$

Here, the equation (3.33) is derived as compatibility conditions of (3.30) and (3.31). While, being

$$\alpha(\xi_a + \eta) \neq 0, \quad \varphi_t(\xi_a) \neq 0 \ \forall a \in \{1, ..., N\}, \tag{3.35}$$

the equation (3.34) is derived from (3.32) multiplying both sides of it for the finite nonzero ratio $\alpha(\xi_a + \eta)/\varphi_t(\xi_a)$ and by using (3.30). $\square$

### 3.2.2 Completeness of Bethe Ansatz solutions by SoV for non-invertible twist

As detailed in the introduction, Nested Algebraic and Analytic Bethe Ansatz have been used to study the spectrum of the model associated to the fundamental representation of the $gl_{\mathcal{M}|\mathcal{N}}$ Yang-Baxter superalgebra. For the fundamental representation of the $gl_{1|2}$ Yang-Baxter superalgebra associated to a simple and diagonalizable twist matrix $K$, let us recall here the form of the Bethe Ansatz equations [3, 134–136]:

$$k_1 Q_2(\lambda_j)a(\lambda_j) = k_2 d(\lambda_j)Q_2(\lambda_j + \eta), \tag{3.36}$$

$$k_2 Q_2(\mu_j + \eta)Q_1(\mu_j - \eta) = -k_3 Q_2(\mu_j - \eta)Q_1(\mu_j), \tag{3.37}$$

where

$$Q_1(\lambda) = \prod_{l=1}^{L}(\lambda - \lambda_l), \quad Q_2(\lambda) = \prod_{m=1}^{M}(\lambda - \mu_m), \tag{3.38}$$

and the Bethe Ansatz form of the transfer matrix eigenvalue

$$t_1(\lambda|\{\lambda_{j \leq L}\}, \{\mu_{h \leq M}\}) = \Lambda_1(\lambda) - \Lambda_2(\lambda) - \Lambda_3(\lambda), \tag{3.39}$$

defined by

$$\Lambda_1(\lambda) = a_1(\lambda)\frac{Q_1(\lambda - \eta)}{Q_1(\lambda)}, \tag{3.40}$$

$$\Lambda_2(\lambda) = a_2(\lambda)\frac{Q_1(\lambda - \eta)Q_2(\lambda + \eta)}{Q_1(\lambda)Q_2(\lambda)}, \tag{3.41}$$

$$\Lambda_3(\lambda) = a_3(\lambda)\frac{Q_2(\lambda - \eta)}{Q_2(\lambda)}, \tag{3.42}$$

where

$$a_1(\lambda) = k_1 a(\lambda), \tag{3.43}$$

$$\frac{a_2(\lambda)}{k_2} = \frac{a_3(\lambda)}{k_3} = d(\lambda). \tag{3.44}$$

It is worth to observe that being

$$t_1(\lambda|\{\lambda_{j\leq L}\},\{\mu_{h\leq M}\}) = k_1 a(\lambda)\frac{Q_1(\lambda-\eta)}{Q_1(\lambda)} - d(\lambda)\left(k_2\frac{Q_1(\lambda-\eta)Q_2(\lambda+\eta)}{Q_1(\lambda)Q_2(\lambda)} + k_3\frac{Q_2(\lambda-\eta)}{Q_2(\lambda)}\right),$$
(3.45)

under the following *pair-wise distinct* conditions

$$\lambda_l \neq \lambda_m, \ \mu_p \neq \mu_q, \ \mu_q \neq \lambda_m, \ \forall l \neq m \in \{1,...,L\}, \ p \neq q \in \{1,...,M\},$$
(3.46)

it follows that the function $t_1(\lambda|\{\lambda_{j\leq L}\},\{\mu_{h\leq M}\})$ has only apparent simple poles in the $\lambda_{j\leq L}$ and $\mu_{h\leq M}$. The regularity of $t_1(\lambda|\{\lambda_{j\leq L}\},\{\mu_{h\leq M}\})$ for $\lambda = \lambda_{j\leq L}$ is implied by the Bethe equation (3.36) while the regularity of $t_1(\lambda|\{\lambda_{j\leq L}\},\{\mu_{h\leq M}\})$ for $\lambda = \mu_{j\leq M}$ is implied by the Bethe equation (3.37). Hence[22] $t_1(\lambda|\{\lambda_{j\leq L}\},\{\mu_{h\leq M}\})$ is a polynomial of degree N with the correct asymptotic for a transfer matrix eigenvalue, i.e. it holds:

$$\lim_{\lambda\to\infty} \lambda^{-N} t_1(\lambda|\{\lambda_{j\leq L}\},\{\mu_{h\leq M}\}) = k_1 - (k_2 + k_3).$$
(3.47)

So that the above ansatz is indeed consistent with the analytic properties enjoyed by the transfer matrix eigenvalues. Now, we show that the Bethe ansatz solutions are complete as a corollary of the completeness of the derived quantum spectral curve in the SoV framework, for the class of representations considered in this section. More precisely it holds the next

**Corollary 3.1.** *Let us consider the class of fundamental representations of the $gl_{1|2}$-graded Yang-Baxter algebra associated to non-invertible but simple spectrum $\hat{K}$ twist matrices, with eigenvalues satisfying (3.13). Then, for almost any values of the inhomogeneities, $t_1(\lambda)$ is an eigenvalue of the transfer matrix $T_1^{(\hat{K})}(\lambda|\eta)$ iff. there exists a solution $\{\{\lambda_{j\leq L}\},\{\mu_{h\leq M}\}\}$ to the Bethe Ansatz equations (3.36) and (3.37) such that $t_1(\lambda) \equiv t_1(\lambda|\{\lambda_{j\leq L}\},\{\mu_{h\leq M}\})$, i.e. $t_1(\lambda)$ has the Bethe Ansatz form (3.45) associated to the solutions $\{\lambda_{j\leq L}\}, \{\mu_{h\leq M}\}$. Moreover, for any $t_1(\lambda) \in \Sigma_{T^{(\hat{K})}}$ the associated Bethe Ansatz solution $\{\{\lambda_{j\leq L}\},\{\mu_{h\leq M}\}\}$ is unique and satisfies the admissibility conditions:*

$$\{\lambda_{j\leq L}\} \subset \{\xi_{j\leq N}\}, \ \ \{\mu_{h\leq M}\} \cap \{\{\xi_{j\leq N}\} \cup \{\xi_{j\leq N} + \eta\}\} = \emptyset.$$
(3.48)

*Proof.* This corollary directly follows from our previous theorem. The proof is done pointing out the consequences of the special form of the fusion equations for these representations associated to these non-invertible twists. In particular, from the fusion equations (3.33) and (3.34), which have to be satisfied by all the transfer matrix eigenvalues, we derive the following equation on the second transfer matrix eigenvalues only:

$$t_2^{(\hat{K})}(\xi_a)t_2^{(\hat{K})}(\xi_a + \eta) = 0.$$
(3.49)

Being by definition $t_2^{(\hat{K})}(\lambda)$ a degree 2N polynomial in $\lambda$, zero in the points $\xi_a - \eta$ for any $a \in \{1,...,N\}$, it follows that a solution to (3.49) can be obtained iff for any $a \in \{1,...,N\}$ there exists a unique $h_a \in \{-1,0\}$ such that:

$$t_{2,\mathbf{h}}^{(\hat{K})}(\lambda) = k_2 k_3 \prod_{a=1}^{N}(\lambda - \xi_a + \eta)(\lambda - \xi_a^{(h_a)}).$$
(3.50)

So we have that the system (3.49) has exactly $2^N$ distinct solutions associated to the $2^N$ distinct N-uplet $\mathbf{h} = \{h_{1\leq n\leq N}\}$ in $\{-1,0\}^N$. Now for any fixed $\mathbf{h} \in \{-1,0\}^N$ we can define a permutation $\pi_{\mathbf{h}} \in S_N$ and a non-negative integer $m_{\mathbf{h}} \leq N$ such that:

$$h_{\pi_{\mathbf{h}}(a)} = 0, \ \ \forall a \in \{1,...,m_{\mathbf{h}}\} \ \text{ and } \ h_{\pi_{\mathbf{h}}(a)} = -1, \ \ \forall a \in \{m_{\mathbf{h}}+1,...,N\}.$$
(3.51)

---

[22]One should also ask for some condition like $\{\mu_{h\leq M}\} \cap \{\xi_{j\leq N}\} = \emptyset$ from which the Bethe equation (3.36) implies $\{\lambda_{h\leq L}\} \cap \{\xi_{j\leq N}\} = \emptyset$ unless $k_1 = 0$.

It is easy to remark now that fixed $\mathbf{h} \in \{-1,0\}^N$ then (3.33), for $a \in \{1, ..., m_{\mathbf{h}}\}$, and (3.34) are satisfied iff it holds:

$$t_{1,\mathbf{h}}^{(\hat{K})}(\xi_{\pi_{\mathbf{h}}(a)}) = 0, \quad \forall a \in \{1, ..., m_{\mathbf{h}}\}. \tag{3.52}$$

Indeed, if this is not the case for a given $b \in \{1, ..., m_{\mathbf{h}}\}$, then (3.34) implies $t_{2,\mathbf{h}}^{(\hat{K})}(\xi_{h_{\pi_{\mathbf{h}}(b)}} + \eta) = 0$ which is not compatible with our choice of $t_{2,\mathbf{h}}^{(\hat{K})}(\lambda)$. So, for any fixed $\mathbf{h} \in \{0,-1\}^N$, we have that the eigenvalues of the transfer matrix have the following form:

$$t_{1,\mathbf{h}}^{(\hat{K})}(\lambda) = \bar{t}_{1,\mathbf{h}}^{(\hat{K})}(\lambda) \prod_{a=1}^{m_{\mathbf{h}}} (\lambda - \xi_{\pi_{\mathbf{h}}(a)}), \tag{3.53}$$

where $\bar{t}_{1,\mathbf{h}}^{(\hat{K})}(\lambda)$ is a degree $N - m_{\mathbf{h}}$ polynomial in $\lambda$, and the function $\varphi_{t,\mathbf{h}}(\lambda)$ associated by the quantum spectral curve to the eigenvalue $t_{1,\mathbf{h}}^{(\hat{K})}(\lambda)$ has the form:

$$\varphi_{t,\mathbf{h}}(\lambda) = \bar{\varphi}_{t,\mathbf{h}}(\lambda) \prod_{a=1}^{m_{\mathbf{h}}} (\lambda - \xi_{\pi_{\mathbf{h}}(a)} - \eta), \tag{3.54}$$

where $\bar{\varphi}_{t,\mathbf{h}}(\lambda)$ is of degree $M - m_{\mathbf{h}} \le M$ polynomial in $\lambda$. Then, simplifying common prefactors, the quantum spectral curve rewrite as it follows:

$$\bar{t}_{1,\mathbf{h}}^{(\hat{K})}(\lambda) \bar{\varphi}_{t,\mathbf{h}}(\lambda) = \bar{\alpha} \, \bar{d}_{\mathbf{h}}(\lambda - \eta) \bar{\varphi}_{t,\mathbf{h}}(\lambda + \eta) + \frac{k_2 k_3}{\bar{\alpha}} \bar{d}_{\mathbf{h}}(\lambda) \bar{\varphi}_{t,\mathbf{h}}(\lambda - \eta), \tag{3.55}$$

where we have defined:

$$\bar{d}_{\mathbf{h}}(\lambda) = \prod_{a=m_{\mathbf{h}}+1}^{N} (\lambda - \xi_{\pi_{\mathbf{h}}(a)}). \tag{3.56}$$

So, once we chose $\bar{\alpha} = -k_2$, we get the following representation of the transfer matrix eigenvalue:

$$t_{1,\mathbf{h}}^{(\hat{K})}(\lambda) = -\prod_{a=1}^{m_{\mathbf{h}}} (\lambda - \xi_{\pi_{\mathbf{h}}(a)}) \frac{k_2 \, \bar{d}_{\mathbf{h}}(\lambda - \eta) \bar{\varphi}_{t,\mathbf{h}}(\lambda + \eta) + k_3 \bar{d}_{\mathbf{h}}(\lambda) \bar{\varphi}_{t,\mathbf{h}}(\lambda - \eta)}{\bar{\varphi}_{t,\mathbf{h}}(\lambda)}. \tag{3.57}$$

It is now trivial to verify that this coincides with the Bethe ansatz form (3.45) for $k_1 = 0$, once we fix:

$$Q_1^{\left(t_{1,\mathbf{h}}^{(\hat{K})}\right)}(\lambda) \equiv \bar{d}_{\mathbf{h}}(\lambda), \quad Q_2^{\left(t_{1,\mathbf{h}}^{(\hat{K})}\right)}(\lambda) \equiv \bar{\varphi}_{t,\mathbf{h}}(\lambda). \tag{3.58}$$

Clearly by definition $Q_1^{\left(t_{1,\mathbf{h}}^{(\hat{K})}\right)}(\lambda)$ and $Q_2^{\left(t_{1,\mathbf{h}}^{(\hat{K})}\right)}(\lambda)$ are solutions of the Bethe Ansatz equations (3.36) and (3.37) and their roots satisfy the conditions (3.48). $\square$

*Remark* 3.4. It is worth to point out that the above set of Bethe Ansatz solutions indeed satisfies also the pair-wise distinct conditions (3.46). Indeed, from the proof of the previous corollary, we know that for any fixed $\mathbf{h} \in \{-1,0\}^N$, there are $2^{N-m_{\mathbf{h}}}$ eigenvalues of the transfer matrix of the form (3.57) associated to as many polynomials $\bar{\varphi}_{t,\mathbf{h}}(\lambda)$ of degree $M \le N - m_{\mathbf{h}}$ in $\lambda$. For any fixed $\mathbf{h} \in \{-1,0\}^N$, these are solutions to (3.37) which coincide with the system of Bethe Ansatz equations associated to an inhomogeneous XXX spin 1/2 quantum chain with $N - m_{\mathbf{h}}$ quantum sites, with inhomogeneities $\xi_{\pi_{\mathbf{h}}(a)}$ for $a \in \{m_{\mathbf{h}}+1, ..., N\}$ and parameter $-\eta$. Then, to these Bethe Ansatz solutions apply the results of the paper [248] which implies the *pair-wise distinct* conditions

$$\mu_p \ne \mu_q, \ \forall \ p \ne q \in \{1, ..., M\}, \tag{3.59}$$

which together with the already proven (3.48) imply in particular (3.46).

# 4 Separation of variables basis for inhomogeneous Hubbard model

## 4.1 The inhomogeneous Hubbard model

The 1+1 dimensional Hubbard model is integrable in the quantum inverse scattering approach with respect to the Shastry's $R$-matrix, which contains as a special case the Lax operator of the Hubbard model [120–122]. In order to introduce them let us start defining the following functions:

$$h(\lambda, \eta) : \sinh 2h(\lambda, \eta) = \frac{i\eta}{2} \sin 2\lambda, \quad \Lambda(\lambda) = -i \cot g(2\lambda) \cosh(2h(\lambda, \eta)), \tag{4.1}$$

here, we use the notation $\eta = -2iU$ with the parameter $U$, the coupling of the Hubbard model, as it plays a similar role as the parameter $\eta$ in the XXX model from the point of view of the Bethe equations. In the following we omit the $\eta$ dependence in $h(\lambda, \eta)$ if not required. Then the Shastry's $R$-matrix reads:

$$R_{12,34}(\lambda|\mu) = I_{12}(h(\lambda))I_{34}(h(\mu))\hat{R}_{12,34}(\lambda|\mu)I_{12}(-h(\lambda))I_{34}(-h(\mu)), \tag{4.2}$$

where

$$\hat{R}_{12,34}(\lambda|\mu) = R_{1,3}(\lambda-\mu)R_{2,4}(\lambda-\mu) - \frac{\sin(\lambda-\mu)}{\sin(\lambda+\mu)}\tanh(h(\lambda)+h(\mu))R_{1,3}(\lambda+\mu)\sigma_1^y R_{2,4}(\lambda+\mu)\sigma_2^y, \tag{4.3}$$

and

$$R_{a,b}(\lambda) = \begin{pmatrix} \cos\lambda & 0 & 0 & 0 \\ 0 & \sin\lambda & 1 & 0 \\ 0 & 1 & \sin\lambda & 0 \\ 0 & 0 & 0 & \cos\lambda \end{pmatrix} \in \text{End}(V_a \otimes V_b), \tag{4.4}$$

where $V_a \cong V_b \cong \mathbb{C}^2$ and we have defined:

$$I_{1,2}(h) = \cosh h/2 + \sigma_1^y \otimes \sigma_2^y \sinh h/2 = \exp(\sigma_1^y \sigma_2^y h/2), \tag{4.5}$$

which satisfies the Yang-Baxter equation:

$$R_{A,B}(\lambda|\mu)R_{A,C}(\lambda|\xi)R_{B,C}(\mu|\xi) = R_{B,C}(\mu|\xi)R_{A,C}(\lambda|\xi)R_{A,B}(\lambda|\mu) \in \text{End}(V_A \otimes V_B \otimes V_C), \tag{4.6}$$

where we have used the capital Latin letters to represent a couple of integers, for example $A = (1,2), \ B = (3,4), \ C = (5,6)$, meaning that:

$$V_A = V_1 \otimes V_2 \cong \mathbb{C}^4, \ V_B = V_3 \otimes V_4 \cong \mathbb{C}^4, \ V_C = V_5 \otimes V_6 \cong \mathbb{C}^4. \tag{4.7}$$

This $R$-matrix satisfies the following properties:

$$R_{A,B}(\lambda|\lambda) = P_{1,3}P_{2,4}, \tag{4.8}$$

where $P_{i,j}$ are the permutation operators on the two-dimensional spaces $V_i \cong V_j \cong \mathbb{C}^2$, moreover, it holds:

$$R_{A,B}(\lambda|0) = \frac{L_{A,B}(\lambda)}{\cosh h(\lambda)}, \quad R_{A,B}(0|\lambda) = \frac{L_{A,B}(-\lambda)}{\cosh h(\lambda)}, \tag{4.9}$$

where $L_{A,B}(\lambda)$ is the Lax operator for the homogeneous Hubbard model:

$$L_{A,B}(\lambda) = I_{12}(h(\lambda))R_{1,3}(\lambda)R_{2,4}(\lambda)I_{12}(h(\lambda)). \tag{4.10}$$

We have the following unitarity property:

$$R_{A,B}(\lambda|\mu)R_{B,A}(\mu|\lambda) = \cos^2(\lambda-\mu)(\cos^2(\lambda-\mu)-\cos^2(\lambda+\mu)\tanh(h(\lambda)-h(\mu))), \quad (4.11)$$

and crossing unitarity relations:

$$R_{A,B}^{-1}(\lambda|\mu) \propto \sigma_1^y \otimes \sigma_2^y R_{A,B}^{t_A}(\lambda-\eta|\mu)\sigma_1^y \otimes \sigma_2^y, \quad (4.12)$$

$$R_{A,B}^{-1}(\lambda|\mu) \propto \sigma_3^y \otimes \sigma_4^y R_{A,B}^{t_B}(\lambda|\mu+\eta)\sigma_3^y \otimes \sigma_4^y. \quad (4.13)$$

This $R$-matrix satisfies the following symmetry properties, i.e. scalar Yang-Baxter equation:

$$R_{A,B}(\lambda|\mu)K_A K_B = K_B K_A R_{A,B}(\lambda|\mu) \in \text{End}(V_A \otimes V_B), \quad (4.14)$$

where $K \in \text{End}(V \cong \mathbb{C}^4)$ is any $4 \times 4$ matrix of the form:

$$K(a,\alpha,\beta,\gamma) = \delta_{a,1}\begin{pmatrix} \alpha & 0 & 0 & 0 \\ 0 & \beta & 0 & 0 \\ 0 & 0 & \gamma & 0 \\ 0 & 0 & 0 & \beta\gamma/\alpha \end{pmatrix} + \delta_{a,2}\begin{pmatrix} \alpha & 0 & 0 & 0 \\ 0 & 0 & \beta & 0 \\ 0 & \gamma & 0 & 0 \\ 0 & 0 & 0 & \beta\gamma/\alpha \end{pmatrix}$$
$$+ \delta_{a,3}\begin{pmatrix} 0 & 0 & 0 & \alpha \\ 0 & \beta & 0 & 0 \\ 0 & 0 & \gamma & 0 \\ \beta\gamma/\alpha & 0 & 0 & 0 \end{pmatrix} + \delta_{a,4}\begin{pmatrix} 0 & 0 & 0 & \alpha \\ 0 & 0 & \beta & 0 \\ 0 & \gamma & 0 & 0 \\ \beta\gamma/\alpha & 0 & 0 & 0 \end{pmatrix}, \quad (4.15)$$

where $\alpha,\beta$ and $\gamma$ are generic complex values. Note that $K(1,\alpha,\beta,\gamma)$ is simple for generic different values of $\alpha,\beta,\gamma$ satisfying $\beta\gamma/\alpha \neq \alpha,\beta,\gamma$. Being $\{\alpha,\beta\gamma/\alpha,\sqrt{\beta\gamma},-\sqrt{\beta\gamma}\}$ the eigenvalues of $K(2,\alpha,\beta,\gamma)$, then $K(2,\alpha,\beta,\gamma)$ is simple for generic nonzero values of $\alpha,\beta,\gamma$ satisfying $\beta\gamma \neq \alpha^2$. Being $\{\beta,\gamma,\sqrt{\beta\gamma},-\sqrt{\beta\gamma}\}$ the eigenvalues of $K(3,\alpha,\beta,\gamma)$, then $K(3,\alpha,\beta,\gamma)$ is simple for generic different and nonzero values of $\beta,\gamma$. The matrix $K(4,\alpha,\beta,\gamma)$ is instead degenerate being $\{\sqrt{\beta\gamma},-\sqrt{\beta\gamma}\}$ its eigenvalues.

We can define the following monodromy matrix:

$$M_A^{(K)}(\lambda) \equiv K_A R_{A,A_N}(\lambda|\xi_N)\cdots R_{A,A_1}(\lambda|\xi_1) \in \text{End}(V_A \otimes \mathcal{H}), \quad (4.16)$$

where $\mathcal{H} = \bigotimes_{n=1}^N V_{A_n}$, $V_{A_n} \cong \mathbb{C}^4$. Then the transfer matrix:

$$T^{(K)}(\lambda) \equiv tr_A M_A^{(K)}(\lambda), \quad (4.17)$$

defines a one-parameter family of commuting operators.

## 4.2 Our SoV covector basis

The general Proposition 2.4 and 2.5 of [1] for the construction of the SoV covector basis and the diagonalizability and simplicity of the transfer matrix spectrum can be adapted to the inhomogeneous Hubbard model. Let us denote with $K_J(a,\alpha,\beta,\gamma)$ the diagonal form of the matrix $K(a,\alpha,\beta,\gamma)$ and $W_K$ the invertible matrix defining the change of basis to it:

$$K = W_K K_J W_K^{-1}, \quad (4.18)$$

clearly $W_K$ is the identity for $a = 1$, then the following theorem holds:

**Theorem 4.1.** *For almost any choice of the inhomogeneities under the condition* (2.81) *and of the twist matrix* $K(a, \alpha, \beta, \gamma)$*, for* $a = 1, 2, 3$*, the Hubbard transfer matrix* $T^{(K)}(\lambda)$ *is diagonalizable and with simple spectrum and the following set of covectors:*

$$\langle h_1, ..., h_N| \equiv \langle S| \prod_{n=1}^{N} (T^{(K)}(\xi_n))^{h_n} \text{ for any } \{h_1, ..., h_N\} \in \{0, 1, 2, 3\}^{\otimes N}, \quad (4.19)$$

*forms a covector basis of* $\mathscr{H}$*, for almost any choice of* $\langle S|$*. In particular, we can take the state* $\langle S|$ *of the following tensor product form:*

$$\langle S| = \bigotimes_{a=1}^{N} (x, y, z, w)_a \Gamma_W^{-1}, \quad \Gamma_W = \bigotimes_{a=1}^{N} W_{K,a}, \quad (4.20)$$

*simply asking* $x y z w \neq 0$.

*Proof.* We have just to remark that also in this case the following identity holds:

$$T^{(K)}(\xi_n) = R_{A_n, A_{n-1}}(\xi_n | \xi_{n-1}) \cdots R_{A_n, A_1}(\xi_n | \xi_1) K_{A_n} R_{A_n, A_N}(\xi_n | \xi_N) \cdots R_{A_n, A_{n+1}}(\xi_n | \xi_{n+1}). \quad (4.21)$$

Let us now point out that $e^{h(\lambda, \eta)}$ is an algebraic function of order two in $\eta$ and $e^\lambda$. Then the determinant of the matrix whose rows are the elements of these covectors in the elementary basis is also an algebraic function of $\{e^{\xi_m}\}_{m \in \{1, ..., N\}}$ and $\eta$. So that showing that this determinant is nonzero for a specific value of $\eta$ one can prove that it is nonzero for almost any value of $\eta$ and of the others parameters, i.e. the inhomogeneities satisfying (2.81) and the parameters $\alpha, \beta, \gamma$ of the twist matrix, for $a = 1, 2, 3$. We can study for example the case $\eta = 0$. In this case $h(\lambda, \eta)$ has the following two different determinations:

$$h(\lambda, \eta = 0) = 0, i\pi/2 \bmod i\pi. \quad (4.22)$$

Note that in both the cases, we have that it holds:

$$\tanh(h(\lambda, 0) + h(\mu, 0)) = 0, \quad (4.23)$$

so that the Shastry's $R$-matrix reduces to the tensor product of two XX $R$-matrix, i.e. it holds:

$$R_{A \equiv (1,2), B \equiv (3,4)}(\lambda | \mu)_{\eta=0} = R_{1,3}(\lambda - \mu) R_{2,4}(\lambda - \mu). \quad (4.24)$$

In turn this implies that:

$$\lim_{\lambda \to -i\infty} e^{-i\lambda} R_{A \equiv (1,2), B \equiv (3,4)}(\lambda | \mu)_{\eta=0} = (e^{-i\mu}/4) \mathbb{I}_{V_A \otimes V_B}, \quad (4.25)$$

so that we can repeat the same type of proof of the general Proposition 2.4 of [1] to show that for a covector $\langle S|$, of the above tensor product form, the determinant of the full matrix factorizes in the product of the determinants of $4 \times 4$ matrices which are nonzero due to the simplicity of the spectrum of the matrix $K$. This already implies the $w$-simplicity of the transfer matrix $T^{(K)}(\lambda)$ then in the case $\eta = 0$ we can prove the non-orthogonality condition:

$$\langle t|t \rangle \neq 0, \quad (4.26)$$

for any transfer matrix eigenvector by the same argument developed in general Proposition 2.5 of [1], which implies the diagonalizability and simplicity of the transfer matrix spectrum for $\eta = 0$ and so for almost any value of $\eta$ and of the others parameters of the representation. $\quad \square$

Let us briefly comment about the consequences of the existence of such a basis. The first important point to stress is that whenever we have an eigenvalue for the transfer matrix, we can write the corresponding eigenvector in the above basis. It means that if we compute a set of solutions to the Nested Bethe Ansatz equations we can immediately write the transfer matrix eigenvalue and hence the corresponding eigenvector; in particular it will be a true eigenvector as soon as it is non zero. This could be of great use in practice when dealing with finite chains with a number of sites greater than the values accessible by direct diagonalization. In particular, scalar products and form factors could become accessible, at least numerically, from this procedure. For using the above basis on a more fundamental, analytical level, one needs to obtain the complete set of fusion relations that lead to the full closure relations enabling to compute the action of the transfer matrix in the SoV basis (see the discussion on this point given in [1, 159]). This should lead to the full characterization of the spectrum. These fusion relations being rather involved for the Hubbard model, due in particular to the intricate dependence on spectral parameters [132], we will come back to this question in a future publication. Let us nevertheless anticipate that the results obtained for the $gl_{\mathscr{M}|\mathscr{N}}$ case will be of direct importance when dealing with the Hubbard model.

# Acknowledgements

J.M.M. and G.N. are supported by CNRS and ENS de Lyon. L.V. is supported by École Polytechnique and ENS de Lyon. The authors would like to thank F. Delduc, M. Magro and H. Samtleben for their availability and for interesting discussions.

# A Compatibility of SoV and Bethe Ansatz framework

In this appendix, we verify how the results obtained in the Nested Algebraic and Analytic Bethe Ansatz framework for the $gl_{1|2}$ Yang-Baxter superalgebra are compatible with the conjectured spectrum characterization in the SoV basis. This analysis is done in the fundamental representations of the $gl_{1|2}$ Yang-Baxter superalgebra associated to generic diagonalizable and simple spectrum twist matrices.

## A.1 Compatibility conditions for higher transfer matrix eigenvalues

Here we use the Bethe Ansatz form (3.39) of the transfer matrix eigenvalues together with the Bethe ansatz equations (3.36) and (3.37) to describe the eigenvalues of the higher transfer matrices in order to verify that they satisfy both the null out-boundary (3.7) and the inner-boundary (3.4) conditions. Under these hypothesis, we get the following lemma:

**Lemma A.1.** *The eigenvalues of the higher transfer matrices admit the following representation in terms of the $\Lambda_i(\lambda)$ functions:*

$$t_2(\lambda) = \Lambda_1(\lambda)(k_1 t_1(\lambda + \eta) + k_3 k_2 d(\lambda))/k_1 \tag{A.1}$$
$$= \Lambda_1(\lambda)(\Lambda_1(\lambda + \eta) - \Lambda_2(\lambda + \eta) - \Lambda_3(\lambda + \eta) + k_3 k_2 d(\lambda)/k_1), \tag{A.2}$$

*and*

$$t_{n+1}(\lambda) = \Lambda_1(\lambda) t_n(\lambda + \eta) \quad \forall n \geq 2. \tag{A.3}$$

*Proof.* We have already observed that due to the Corollary 2.1 the eigenvalues of the higher transfer matrices admit the interpolation formulae (2.108) in terms of the transfer matrix

eigenvalue $t_1(\lambda)$. Equivalently, given $t_1(\lambda)$ an eigenvalue of the transfer matrix then those of the higher transfer matrices are of the form

$$t_n(\lambda) = \prod_{r=1}^{n-1} d(\lambda + r\eta)\tilde{t}_n(\lambda) \quad \forall n \geq 2, \tag{A.4}$$

where $\tilde{t}_n(\lambda)$ are degree N polynomials in $\lambda$, fixed uniquely by the recursive equations:

$$t_2(\xi_a) = t_1(\xi_a)t_1(\xi_a + \eta), \tag{A.5}$$

$$t_{n+1}(\xi_a) = t_1(\xi_a)t_n(\xi_a + \eta), \tag{A.6}$$

and the known asymptotics:

$$\lim_{\lambda \to \infty} \lambda^{-N}t_n(\lambda) = T^{(K)}_{\infty,n} = k_1^{n-2}(k_1 - k_3)(k_1 - k_2), \quad \forall n \geq 2. \tag{A.7}$$

So to prove the above Bethe Ansatz form for the higher transfer matrix eigenvalues we have just to verify these conditions. Concerning the asymptotic behaviour, from the r.h.s. of formula (A.1) and (A.3) we get:

$$k_1(k_1 - k_3 - k_2) + k_3k_2 = (k_1 - k_3)(k_1 - k_2) \tag{A.8}$$

$$= \left((\mathrm{str}K)^2 + \left(\mathrm{str}K^2\right)\right)/2 = T^{(K)}_{\infty,2}, \tag{A.9}$$

and

$$\lim_{\lambda \to \infty} \lambda^{-N}t_n(\lambda) = k_1^{n-2}T^{(K)}_{\infty,2} = T^{(K)}_{\infty,n}, \tag{A.10}$$

so they are satisfied. So we are left with the proof of the fusion properties. It is easy to remark that by the definition of the $\Lambda_i(\lambda)$ it follows that the $t_n(\lambda)$ indeed factorize the coefficients $\prod_{r=1}^{n-1} d(\lambda + r\eta)$, for $n \geq 2$. Let us now show that

$$\tilde{t}_2(\lambda) = \frac{Q_1(\lambda - \eta)}{Q_1(\lambda)}(k_1 t_1(\lambda + \eta) + k_3 k_2 d(\lambda)), \tag{A.11}$$

is indeed a degree $N$ polynomial in $\lambda$. This is the case iff the residues of this expression in the zeroes of $Q_1(\lambda)$ are vanishing, namely iff the following identities hold:

$$t_1(\lambda_j + \eta) = -\frac{k_3 k_2}{k_1}d(\lambda_j) \text{ for any } j \in \{1, ..., L\}, \tag{A.12}$$

and this is the case thanks to the Bethe equation (3.36) in $\lambda_j$ being:

$$t_1(\lambda_j + \eta) = -\Lambda_3(\lambda_j + \eta) = -k_3 a(\lambda_j)\frac{Q_2(\lambda_j)}{Q_2(\lambda_j + \eta)}. \tag{A.13}$$

Similarly, we have that

$$\tilde{t}_n(\lambda) = k_1 \frac{Q_1(\lambda - \eta)}{Q_1(\lambda)}\tilde{t}_{n-1}(\lambda + \eta) \tag{A.14}$$

$$= k_1^{n-2}\frac{Q_1(\lambda - \eta)}{Q_1(\lambda + (n-3)\eta)}\tilde{t}_2(\lambda + (n-2)\eta) \tag{A.15}$$

$$= k_1^{n-2}\frac{Q_1(\lambda - \eta)}{Q_1(\lambda + (n-2)\eta)}(k_1 t_1(\lambda + \lambda + (n-1)\eta) + k_3 k_2 d(\lambda + (n-2)\eta)), \tag{A.16}$$

which is a degree $N$ polynomial in $\lambda$ due to the identity (3.36).

So to show that the $t_n(\lambda)$ satisfy the characterization of the higher eigenvalues, we have just to verify that their values in the inhomogeneities agree with (A.5) and (A.6). Indeed, it holds:

$$
\begin{aligned}
t_2(\xi_a) &= \Lambda_1(\xi_a)(k_1 t_1(\xi_a + \eta) + k_3 k_2 d(\xi_a))/k_1 \\
&= \Lambda_1(\xi_a) t_1(\xi_a + \eta) \\
&= t_1(\xi_a) t_1(\xi_a + \eta),
\end{aligned}
\tag{A.17}
$$

where in the last line we have used the Bethe Ansatz form of $t_1(\lambda)$ and similarly:

$$
t_{n+1}(\xi_a) = \Lambda_1(\xi_a) t_n(\xi_a + \eta) = t_1(\xi_a) t_n(\xi_a + \eta).
\tag{A.18}
$$

$\square$

Here we have explicitly rewritten the eigenvalues form in Bethe Ansatz approach for the higher transfer matrix $T_n^{(\hat{K})}(\lambda)$, by using the fusion we can easily derive those of the others. Now we are interested in showing that these expressions for the higher eigenvalues indeed imply the null out-boundary (3.7) and the inner-boundary (3.4) conditions. Indeed, we have the following lemma:

**Lemma A.2.** *Let us take a Bethe equation solution and associate to it the $t_1(\lambda)$ of the form (3.45), then the higher functions $t_n^{(m)}(\lambda)$ generated from $t_1(\lambda)$ by the fusion equations, i.e. by using (2.57), (2.58), (2.82) and (2.83), satisfy the null out-boundary condition (3.7) and the inner-boundary (3.4).*

*Proof.* By using the result of the previous lemma it is easy to show the following null conditions are satisfied:

$$
t_{3+n}^{(2)}(\lambda) = 0, \quad \forall n \geq 0,
\tag{A.19}
$$

indeed, the condition (A.3) implies:

$$
\Lambda_1(\lambda) = t_{3+n}(\lambda)/t_{2+n}(\lambda + \eta), \quad \forall n \geq 0,
\tag{A.20}
$$

so that, in particular, it holds:

$$
t_{3+n}(\lambda) = (t_{2+n}(\lambda)/t_{1+n}(\lambda + \eta))(t_{2+n}(\lambda + \eta)), \quad \forall n \geq 1,
\tag{A.21}
$$

or equivalently:

$$
t_{2+n}(\lambda) t_{2+n}(\lambda + \eta) = t_{3+n}(\lambda) t_{1+n}(\lambda + \eta), \quad \forall n \geq 1,
\tag{A.22}
$$

which by the fusion equations implies the above null conditions. Similarly, we can derive all the other null out-boundary conditions (3.7).

Let us now show the *inner-boundary* condition, from the formula (A.1) we can write:

$$
\Lambda_1(\lambda) = \frac{t_2(\lambda)}{t_1(\lambda + \eta) + k_3 k_2 d(\lambda)/k_1},
\tag{A.23}
$$

and so

$$
t_3(\lambda) = \frac{k_1 t_2(\lambda) t_2(\lambda + \eta)}{k_1 t_1(\lambda + \eta) + k_3 k_2 d(\lambda)},
\tag{A.24}
$$

which is equivalent to our closure relation:

$$
(k_1 t_1(\lambda + \eta) + k_3 k_2 d(\lambda)) t_3(\lambda) = k_1 t_2(\lambda) t_2(\lambda + \eta),
\tag{A.25}
$$

and taking into account the fusion equation:

$$
t_2^{(2)}(\lambda + \eta) = t_2^{(1)}(\lambda) t_2^{(1)}(\lambda + \eta) - t_1(\lambda + \eta) t_3^{(1)}(\lambda),
\tag{A.26}
$$

we are led to the required identity:

$$k_3 k_2 d(\lambda) t_3^{(1)}(\lambda) = k_1 t_2^{(2)}(\lambda + \eta). \tag{A.27}$$

$\square$

It should be noted that all these relations can be proven in a pure algebraic way using the general constructions of $T$ and $Q$ operators and the various relations they satisfy, as given in [246, 247, 249, 250]. Hence the computations presented here, although quite instructive could be considered merely as consistency checks.

## A.2 On the relation between SoV and Nested Algebraic Bethe Ansatz

Let us consider the fundamental representation of the $gl_{1|2}$ Yang-Baxter superalgebra associated to generic values of the inhomogeneities $\{\xi_{a \leq N}\}$, satisfying the condition (2.81), and of the eigenvalues $k_1$, $k_2$ and $k_3$ of a simple and diagonalizable twist matrix $K$. Note that in this case the transfer matrix is similar to the transfer matrix associated to a diagonal twist with entries the eigenvalues $k_1$, $k_2$ and $k_3$ of $K$ to which Nested Algebraic Bethe Ansatz (NABA) directly applies. Therefore, the following discussion on the connection between the SoV description and the NABA can be directly addressed in this diagonal case.

Let us recall that in the NABA framework, given a solution $\{\{\lambda_{j \leq L}\}, \{\mu_{h \leq M}\}\}$ of the Bethe Ansatz equations (3.36) and (3.37) satisfying the pair-wise distinct conditions (3.46), then the associated Bethe Ansatz vector $|t_{\{\lambda_{j \leq L}\}, \{\mu_{h \leq M}\}}\rangle$ is proven to satisfy the identity

$$T_1^{(K)}(\lambda) | t_{\{\lambda_{j \leq L}\}, \{\mu_{h \leq M}\}}^{(NABA)} \rangle = | t_{\{\lambda_{j \leq L}\}, \{\mu_{h \leq M}\}}^{(NABA)} \rangle t_1(\lambda | \{\lambda_{j \leq L}\}, \{\mu_{h \leq M}\}), \tag{A.28}$$

with $t_1(\lambda | \{\lambda_{j \leq L}\}, \{\mu_{h \leq M}\})$ defined in (3.39), so that it is a transfer matrix eigenvector as soon as it is proven to be nonzero. Then, such a Bethe Ansatz vector has in our SoV basis the following characterization:

$$\langle h_1, ..., h_N | t_{\{\lambda_{j \leq L}\}, \{\mu_{h \leq M}\}}^{(NABA)} \rangle = \prod_{n=1}^{N} t_1^{h_n}(\xi_n | \{\lambda_{j \leq L}\}, \{\mu_{h \leq M}\}) \langle S | t_{\{\lambda_{j \leq L}\}, \{\mu_{h \leq M}\}}^{(NABA)} \rangle, \tag{A.29}$$

for any $h_n \in \{0, 1, 2\}$ and $n \in \{0, ..., N\}$. Note that also in the SoV basis the condition that this Bethe vector is nonzero still remains to be verified. This is the case even for the special representations considered in subsection 3.2. Indeed, we have shown that the specific set of solutions to the Bethe Ansatz equations (3.36) and (3.37) introduced in subsection 3.2.2 is complete and the associated eigenvalues $t_1(\lambda | \{\lambda_{j \leq L}\}, \{\mu_{h \leq M}\})$ and eigenvectors $|t_{\{\lambda_{j \leq L}\}, \{\mu_{h \leq M}\}}^{(SoV)}\rangle$ have the form (3.45) and (3.29), i.e.

$$\langle h_1, ..., h_N | t_{\{\lambda_{j \leq L}\}, \{\mu_{h \leq M}\}}^{(SoV)} \rangle = \prod_{n=1}^{N} t_1^{h_n}(\xi_n | \{\lambda_{j \leq L}\}, \{\mu_{h \leq M}\}), \tag{A.30}$$

eigenvectors known to be nonzero by the characterization of the transfer matrix eigenvalues for which there exists at least one N-uplet $h_1, ..., h_N$ leading to a nonzero value of the above SoV wave-function. Nevertheless, this a priori does not allow us to rule out the possibility that:

$$| t_{\{\lambda_{j \leq L}\}, \{\mu_{h \leq M}\}}^{(NABA)} \rangle = 0, \tag{A.31}$$

as we have still to verify that $\langle S | t_{\{\lambda_{j \leq L}\}, \{\mu_{h \leq M}\}}^{(NABA)} \rangle$ is nonzero.

Relying on some already existing results in the literature, we want to present a reasoning that allows to argue that the completeness of the Bethe Ansatz in the SoV framework, for the

special representations of subsection 3.2, indeed implies the completeness for the NABA spectrum description. The reasoning goes as follows. In [1,159] we have shown in general that the SoV characterization of the transfer matrix eigenvectors allows for an Algebraic Bethe Ansatz rewriting on a well defined reference state, see for example section 5 of [159]. Adapting to the current fundamental $gl_3$-representation associated the twist matrix $K' = -\hat{K}$ and $\eta' = -\eta$ the analysis of [163], it can be argued[23] that the $\mathbb{B}$-operator defined in our SoV basis indeed coincides with the one defined by Sklyanin [25]. Then by adapting the results presented in [165], one can deduce that these SoV eigenvectors rewritten in an Algebraic Bethe Ansatz form, in terms of the Sklyanin $B$-operator, in turn coincide (up to nonzero normalization) with Nested Algebraic Bethe Ansatz vectors, associated to the same Bethe Ansatz solutions. If implemented with all details this reasoning shows the completeness of the Nested Algebraic Bethe Ansatz as a consequence of the completeness of the SoV characterization derived in subsection 3.2.2 for the fundamental representations associated to non-invertible but simple spectrum $\hat{K}$ twist matrices, with eigenvalues satisfying (3.13).

It is also worth to comment that once the NABA completeness is derived for these special representations, it can be derived for general $gl_{1|2}$-representations by adapting to them the proof given in [248] for the $gl_2$ fundamental representations associated to general diagonalizable twist. Indeed, one of the main ideas of the proof in [248] is that for a special value of the twist parameter, one can characterize the set of isolated Bethe Ansatz solutions that produce nonzero Bethe vectors, and which is proven to be complete. Then, the results on the completeness of Bethe Ansatz solutions by the SoV approach, derived in subsection 3.2.2, and the above argument on the NABA completeness for these $gl_{1|2}$-representations associated to the twist matrices $\hat{K}$ can be as well the starting point for the proof of completeness by deformation w.r.t. the twist parameters like in [248]. Finally, let us add that relations with [204] would be interesting to explore.

## B  Verification of the Conjecture for the general twists up to 3 sites

Here, we make a verification of our conjecture on the form of the closure relations for the general twisted representation of the $gl_{\mathcal{M}|\mathcal{N}}$ Yang-Baxter superalgebra, in the case $\mathcal{M} = 1$ and $\mathcal{N} = 2$ for small chain representations, i.e. for a chain having up to $\mathsf{N} = 3$ sites. The verification is done in the following way, we impose the closure relation (3.6) in $\mathsf{N}$ pairwise different values[24] of $\lambda$ to the polynomials (2.107) and (2.108) for $n = 1, 2$. This determines a system of $\mathsf{N}$ polynomial equations of order 4 in the $\mathsf{N}$ unknowns which are the values of the polynomial (2.107) in the inhomogeneities. We solve this system of equations by Mathematica and we select the solutions which generate polynomials (2.108) which satisfy the null out-boundary conditions 3.7. Our analysis shows that it is enough to impose 3.7 for $n = m = 0$ to select the correct solutions which generate exactly the $\mathsf{N}^3$ different eigenvalues of the diagonalizable and simple spectrum transfer matrix $T^{(K)}(\lambda)$, obtained by diagonalizing it exactly with Mathematica. For $\mathsf{N} = 1, 2$ the results of both the approaches are analytic and we present them here for the interesting $\mathsf{N} = 2$ case. While for $\mathsf{N} = 3$ we have verified our statements for different values of the parameters, i.e. the inhomogeneity parameters and the three eigenvalues of the twist matrix.

We put $\xi_1 = 0$ without loss of generality to shorten the expressions while leaving free all the others parameters $\xi_2, k_1, k_2, k_3$ and $\eta$. Then the solution of the system of equations obtained by (3.6) plus the null out-boundary conditions (3.7) for $n = m = 0$ leads to the following $2^3$ distinct solutions for the values of the polynomial (2.107) respectively in $\lambda = \xi_2$

---

[23]Note that we have proven this statement for a chain with a small number of quantum sites in [1].

[24]Note that any value can be taken if different from the transfer matrix common zeros.

and $\lambda = \xi_1 = 0$:

$$\{k_1\eta(\eta+\xi_2),\ k_1\eta(\eta-\xi_2)\}, \tag{B.1}$$

$$\{k_2\eta(\eta-\xi_2),\ -k_2\eta(\eta+\xi_2)\}, \tag{B.2}$$

$$\{k_3\eta(\eta-\xi_2),\ -k_3\eta(\eta+\xi_2)\}, \tag{B.3}$$

$$\left\{\frac{\eta}{2}\left((k_1+k_2)\xi_2-\sqrt{4k_1k_2\eta^2+(k_1-k_2)^2\xi_2^2}\right),\ \frac{-\eta}{2}\left((k_1+k_2)\xi_2+\sqrt{4k_1k_2\eta^2+(k_1-k_2)^2\xi_2^2}\right)\right\}, \tag{B.4}$$

$$\left\{\frac{\eta}{2}\left((k_1+k_2)\xi_2+\sqrt{4k_1k_2\eta^2+(k_1-k_2)^2\xi_2^2}\right),\ \frac{-\eta}{2}\left((k_1+k_2)\xi_2-\sqrt{4k_1k_2\eta^2+(k_1-k_2)^2\xi_2^2}\right)\right\}, \tag{B.5}$$

$$\left\{\frac{\eta}{2}\left((k_1+k_3)\xi_2-\sqrt{4k_1k_3\eta^2+(k_1-k_3)^2\xi_2^2}\right),\ \frac{-\eta}{2}\left((k_1+k_3)\xi_2+\sqrt{4k_1k_3\eta^2+(k_1-k_3)^2\xi_2^2}\right)\right\}, \tag{B.6}$$

$$\left\{\frac{\eta}{2}\left((k_1+k_3)\xi_2+\sqrt{4k_1k_3\eta^2+(k_1-k_3)^2\xi_2^2}\right),\ \frac{-\eta}{2}\left((k_1+k_3)\xi_2-\sqrt{4k_1k_3\eta^2+(k_1-k_3)^2\xi_2^2}\right)\right\}, \tag{B.7}$$

$$\left\{\frac{\eta}{2}\left((k_2+k_3)\xi_2-\sqrt{4k_2k_3\eta^2+(k_2-k_3)^2\xi_2^2}\right),\ \frac{-\eta}{2}\left((k_2+k_3)\xi_2+\sqrt{4k_2k_3\eta^2+(k_2-k_3)^2\xi_2^2}\right)\right\}, \tag{B.8}$$

$$\left\{\frac{\eta}{2}\left((k_2+k_3)\xi_2-\sqrt{4k_2k_3\eta^2+(k_2-k_3)^2\xi_2^2}\right),\ \frac{-\eta}{2}\left((k_2+k_3)\xi_2-\sqrt{4k_2k_3\eta^2+(k_2-k_3)^2\xi_2^2}\right)\right\}. \tag{B.9}$$

The values at the points $\xi_2$ and $\xi_1 = 0$ and the asymptotic limit allows to reconstruct the polynomials (2.107). The polynomials constructed in this way can be directly verified to coincide with the eigenvalues of $T^{(K)}(\lambda)$, whose expressions are obtained by diagonalizing $T(\lambda)$ exactly with Mathematica:

$$(\mathrm{str}K)\lambda^2 + \left(2\eta k_1 - (\mathrm{str}K)\xi_2\right)\lambda + k_1\eta(\eta-\xi_2), \tag{B.10}$$

$$(\mathrm{str}K)\lambda^2 + \left(2\eta k_2 - (\mathrm{str}K)\xi_2\right)\lambda - k_2\eta(\eta+\xi_2), \tag{B.11}$$

$$(\mathrm{str}K)\lambda^2 + \left(2\eta k_3 - (\mathrm{str}K)\xi_2\right)\lambda - k_3\eta(\eta+\xi_2), \tag{B.12}$$

$$(\mathrm{str}K)\lambda^2 + \left((k_1+k_2)\eta - (\mathrm{str}K)\xi_2\right)\lambda + \frac{\eta}{2}\left(-(k_1+k_2)\xi_2-\sqrt{4k_1k_2\eta^2+(k_1-k_2)^2\xi_2^2}\right), \tag{B.13}$$

$$(\mathrm{str}K)\lambda^2 + \left((k_1+k_2)\eta - (\mathrm{str}K)\xi_2\right)\lambda + \frac{\eta}{2}\left(-(k_1+k_2)\xi_2+\sqrt{4k_1k_2\eta^2+(k_1-k_2)^2\xi_2^2}\right), \tag{B.14}$$

$$(\mathrm{str}K)\lambda^2 + \left((k_1+k_3)\eta - (\mathrm{str}K)\xi_2\right)\lambda + \frac{\eta}{2}\left(-(k_1+k_3)\xi_2-\sqrt{4k_1k_3\eta^2+(k_1-k_3)^2\xi_2^2}\right), \tag{B.15}$$

$$(\mathrm{str}K)\lambda^2 + \left((k_1+k_3)\eta - (\mathrm{str}K)\xi_2\right)\lambda + \frac{\eta}{2}\left(-(k_1+k_3)\xi_2+\sqrt{4k_1k_3\eta^2+(k_1-k_3)^2\xi_2^2}\right), \tag{B.16}$$

$$(\mathrm{str}K)\lambda^2 + \left((k_2+k_3)\eta - (\mathrm{str}K)\xi_2\right)\lambda + \frac{\eta}{2}\left(-(k_2+k_3)\xi_2-\sqrt{4k_2k_3\eta^2+(k_2-k_3)^2\xi_2^2}\right), \tag{B.17}$$

$$(\mathrm{str}K)\lambda^2 + \left((k_2+k_3)\eta - (\mathrm{str}K)\xi_2\right)\lambda + \frac{\eta}{2}\left(-(k_2+k_3)\xi_2+\sqrt{4k_2k_3\eta^2+(k_2-k_3)^2\xi_2^2}\right). \tag{B.18}$$

## C   Derivation of the inner-boundary condition

One may use the coderivative formalism introduced in [246] and developed in [247] to derive the inner-boundary condition. The coderivative formalism allows to construct the transfer

matrices associated to a given irreducible representation on the auxiliary space by acting on the associated character evaluated at the twist matrix. For a rectangular Young tableau $(a, b)$, we have in our notation

$$\widetilde{T}_b^{(a),K}(\lambda) = \left(\lambda - \xi_1 + \eta\hat{D}\right) \otimes \ldots \otimes \left(\lambda - \xi_{\mathsf{N}} + \eta\hat{D}\right) \chi_b^{(a)}(K). \tag{C.1}$$

Let us take $g = \mathrm{diag}(x_1, \ldots, x_{\mathcal{M}}, y_1, \ldots, y_{\mathcal{N}})$ a diagonal twist. For $k \geq 1$, the characters of the rectangular representations $(a, b)$ which saturate an arm of the fat hook write [196]

$$\chi_{\mathcal{N}+k}^{(\mathcal{M})}(g) = \left(\prod_{i=1}^{\mathcal{M}} x_i^k\right) \prod_{i=1}^{\mathcal{M}} \prod_{j=1}^{\mathcal{N}} (x_i - y_j), \tag{C.2}$$

$$\chi_{\mathcal{N}}^{(\mathcal{M}+k)}(g) = \left(\prod_{j=1}^{\mathcal{N}} (-y_j)^k\right) \prod_{i=1}^{\mathcal{M}} \prod_{j=1}^{\mathcal{N}} (x_i - y_j), \tag{C.3}$$

thus the following relation holds for all $k \geq 1$

$$\chi_{\mathcal{N}+k}^{(\mathcal{M})}(g) = (-1)^{k\mathcal{N}} \mathrm{sdet}(g)^k \chi_{\mathcal{N}}^{(\mathcal{M}+k)}(g), \tag{C.4}$$

where $\mathrm{sdet}(g)$ is the superdeterminant of $g$ defined by

$$\mathrm{sdet}(g) = \frac{\prod_{i=1}^{\mathcal{M}} x_i}{\prod_{j=1}^{\mathcal{N}} y_j}. \tag{C.5}$$

Acting on it with the coderivative $\hat{D}$, we have

$$\hat{D}\,\chi_{\mathcal{N}+k}^{(\mathcal{M})}(g) = (-1)^{k\mathcal{N}} e_{ij} \frac{\partial}{\partial \phi_i^j} \otimes \left(\mathrm{sdet}(e^{\phi \cdot e} g)^k \chi_{\mathcal{N}}^{(\mathcal{M}+k)}(e^{\phi \cdot e} g)\right)\Big|_{\phi=0} \tag{C.6}$$

$$= (-1)^{k\mathcal{N}} \mathrm{sdet}(g)^k e_{ij} \frac{\partial}{\partial \phi_i^j} \otimes \left((1 + k\,\mathrm{str}(\phi \cdot e)) \chi_{\mathcal{N}}^{(\mathcal{M}+k)}(e^{\phi \cdot e} g)\right)\Big|_{\phi=0} \tag{C.7}$$

$$= (-1)^{k\mathcal{N}} \mathrm{sdet}(g)^k \left(\hat{D}\chi_{\mathcal{N}}^{(\mathcal{M}+k)} + k\, e_{ij} \frac{\partial}{\partial \phi_i^j} \otimes \left(\mathrm{str}(\phi \cdot e)\, \chi_{\mathcal{N}}^{(\mathcal{M}+k)}(g)\right)\Big|_{\phi=0}\right) \tag{C.8}$$

$$= (-1)^{k\mathcal{N}} \mathrm{sdet}(g)^k (k + \hat{D}) \chi_{\mathcal{N}}^{(\mathcal{M}+k)}(g). \tag{C.9}$$

Now, acting with $\left(\lambda - \xi_1 + \eta\hat{D}\right) \otimes \ldots \otimes \left(\lambda - \xi_{\mathsf{N}} + \eta\hat{D}\right)$ on (C.4), and putting $g = K$, we thus have

$$\widetilde{T}_{\mathcal{N}+k}^{(M),(K)}(\lambda) = (-1)^{k\mathcal{N}} \mathrm{sdet}(K)^k \; \widetilde{T}_{\mathcal{N}}^{(\mathcal{M}+k),(K)}(\lambda + k\eta). \tag{C.10}$$

Putting $k = 1$, and reintroducing the trivial zeros to recover the $T_b^{(a),(K)}(\lambda)$ matrices, we obtain (2.54).

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
