# Peer review of "Separation of variables bases for integrable $gl_{\mathcal{M}|\mathcal{N}}$ and Hubbard models"

_SciPost Physics, doi:SciPost Phys. 9, 060 (2020)_

## Round 2 · Referee Report · Anonymous (Referee 1) · 2020-5-11

Report

The separation of variables is potentially the most powerful method for investigation the off-shell behaviour of the quantum integrable models. In the paper by J.M.Maillert, G. Nicolli and L. Vignoli the problem of separation of variables is solved for models with SUSY. This case is important for studying the Hubbard model, the case considered in details in the paper, another important application can be provided by AdS/CFT correspondence. This is a highly professional and well-written paper, I do recommend it for publication.

---

## Round 2 · Referee Report · Anonymous (Referee 2) · 2020-6-12

Strengths

1- The paper is very well written, and largely self-contained 2- It contains new relevant results on a topic of renewed interest in the integrable systems community. The potential applications of these methods range from condensed matter to AdS/CFT 3- The bibliography and review of the literature is overall comprehensive and balanced.

Weaknesses

1- I feel that some references should be added to other approaches to the completeness problem, and a short discussion on how they compare to the approach of the authors (more comments on this below). 2-Often, the authors refer to arguments in their previous works to shorten the proofs of theorems. The references are very precise, however occasionally I feel that a few more words of explanation should be offered to help the reader follow the logic (I will point out one example below). 3- The review in the introduction of developments in AdS/CFT integrability is in my view incomplete (more comments below).

Report

The paper deals with the separation of variables method in quantum integrable systems, continuing a series of important develoments in the area in the last years, by the authors and other groups.
The paper addresses the construction of separated variables bases for
supersymmetric rational spin chains at any rank, as well as the Hubbard model. This is done applying a method recently developed for the non-supersymmetric case by two of the authors in [1] , whereas the separated variables basis is built by repeated action of the transfer matrix evaluated at special points on a reference state.
After constructing the basis, the authors use it to prove some spectral properties of the transfer matrix, such as simplicity of the spectrum and diagonalizability (under some conditions on the twists).
They also emphasize the role of a nice and simple quantization condition for the transfer matrix eigenvalues (generalizing the "quantum determinant" of the bosonic case), and conjecture that it gives a complete characterization of the spectrum. They prove this statement for a particular choice of twists and superalgebra, providing an explicit example of the construction.
The paper is very well written, and the results are relevant for developing the SoV strategy to the computation of observables in supersymmetric spin chains and AdS/CFT. I have no doubt in recommending the paper for publication. However, I kindly ask the authors to consider some minor corrections detailed below.

Requested changes

REQUESTS OF CLARIFICATIONS AND BIBLIOGRAPHICAL SUGGESTIONS: 1. When discussing completeness, in particular in the Introduction and section 2.5, I would ask to include a reference to the works on completeness based on the Wronskian QQ relations. In particular, to the best of my knowledge the work math.QA/1303.1578 is considered as a proof of completeness of the Bethe Ansatz for the gl(N) XXX spin chain. I also mention the related recent work math-ph/2004.02865 which presents a proof of completeness for the supersymmetric case (of course, this work appeared after the work of the authors, therefore I leave it up to them whether to include this reference). In general, I would really appreciate seeing a short discussion of links and differences of the authors approach with the QQ relations approach. 2. In the second paragraph on page 6 , when talking about generating states with a single, non-nested B operator evaluated at the zeros of the Q function, I would ask to add a citation to [150], where this property was first discovered and the form of B conjectured for any rank.
3. I would like some points to be clarified in the discussion of AdS/CFT integrability, and I kindly ask the authors to take into accounts the following comments. First, the AdS/CFT S matrix is not equivalent to the Hubbard R-matrix, but to two copies of the latter, multiplied by a (nontrivial) dressing phase. Second, the direct relation between the dilatation operator and the Hubbard Hamiltonian is only valid at weak coupling (and in a special sector). Third, spin chain approaches such as those in [148]-[149] are nowadays known to miss one important part of the result for the AdS/CFT spectrum, the so-called wrapping corrections, and have been surpassed by approaches based on the TBA. Finally, I think the authors should include in this discussion the fact that a full solution for the spectral problem of AdS/CFT has been proposed in hep-th/1305.1939 (this has been tested extensively and is regarded as one of the main results in this area). 4.- The study of the Hubbard model in a paper dealing mostly with supersymmetric spin chains could be motivated more strongly. I believe an explicit link exists since the R matrix of the Hubbard model has a su(2|2) invariance (this is e.g. discussed in reference [132] or in math-ph/1401.7691). It could be useful to make this comment. 5. Since the authors cite the fusion hierarchy for the Hubbard model, I point out to them the paper hep-th/1501.04651 where the hierarchy of functional relations and the related quantization conditions are discussed in the finite temperature case. 6. In the proof of Th. 2.1, when mentioning "special limits" used in the proof, it would help the reader if the authors recalled explicitly the limit of large inhomogeneities. 7. Can the authors elaborate on the name "quantum spectral curve" for eq. (3.26)? 8. I feel it would be better to name the zeros of \phi in (3.25) \mu rather than \lambda, consistently with (3.38) and (3.58). 9. This is a simple suggestion. To make contact with other SoV approaches using the B operator, it may be relevant to remark that (3.29) is a rewriting of the wave functions in terms of Q functions.

SOME TYPOS AND NOTATION ISSUES: 10. In (2.90) since this is a covector, should it belong to V_a^* ? 11. I believe there is a typo in (2.68), namely the signs of the shifts in the arguments of the the T functions should be reversed. 12. For notation consistency, in formula (2.36) should it be M^(K) ? Otherwise, I believe the superscript ^(K) comes in (2.43) with no previous explanation. 13. The notation \xi^(n) is used only in two equations. For clarity I would advise writing the shift explicitly in (3.59), since the definition of this notation is difficult to spot.
14. Some typos: "ortogonality" should read "orthogonality" in a couple of places. Another typo is "antisymetric" on page 7. Finally, a typo in one of the names in ref. [135].
15. Below (3.54), the degree of the polynomial \bar \phi is given as M \leq N- m_h. However for consistency with (3.25) , it looks like it should be M- m_h instead. 16. The footnotes 10 and 11 are identical. 17. A power of \lambda is missing in eq. (A.7) 18. It looks like (B.9) and (B.18) are repetitions of (B.8) and (B.17).

---

## Round 2 · Referee Report · Anonymous (Referee 3) · 2020-7-23

Strengths

1-The paper provides extremely comprehensive, with few exceptions, literature review
2-The authors do a great job in providing crystal clear unambiguous definitions of all objects they study. Given notorious sign issues in supersymmetric systems, this is much appreciated
3-Derivation of the inner boundary condition
4-Reconstruction of fused transfer matrices in terms of the fundamental one

Weaknesses

1- The authors do not attempt to interpret their results in terms of supersymmetric Q-systems although it will be potentially beneficial for several reasons. There is a good chance that various formulae simplify in terms of Baxter Q-functions, and there will be a stronger connection with the AdS/CFT integrable system whose spectrum is concisely encoded in terms of a quantum spectral curve (which is a QQ-system). 2- References to recent developments related to the AdS/CFT spectral problem and supersymmetric Q-systems are missing (some suggestions will be given below). This exaggerates to some extent the role of the Hubbard model in application to the AdS/CFT integrability (Hubbard model works in the large volume asymptotic regime only). 3- Although the authors did a great job to make their article detailed and easily understandable, it would be appreciated to have a bit more of cross-references across the text, and occasionally replace phrases like "previous theorem" and "our corollary" with more exact "theorem 2.1" etc. This will make the paper more accessible to those readers who want to understand one particular statement instead of reading the entire article. Also, there are small issues with English grammar, I would suggest proofreading the manuscript.

Report

Separation of variables is an important tool in studies of integrable systems. This article is one of the first works devoted to the development of this tool for the case of supersymmetric systems.

The article offers two main results. The first result is the construction of an SoV basis for the case of twisted $GL(M|N)$ spin chains in fundamental representations and the Hubbard model. The idea is to systematically act with the fundamental transfer matrix on a reference state. While it is a rather straightforward generalisation of the proposal and proofs in [1] (by two of the authors), it was important to explicitly show whether and how it works in the presence of supersymmetry. The second result is to encode the spectrum of the model in terms of certain relations that are imposed directly on the fundamental transfer matrix. This result is conjectured in the general situation and proven in some particular cases (where also completeness is checked/proven for generic values of inhomogeneities).

Although a lot of work is still required in the SoV study for supersymmetric systems, this article provides one of the first stepping stones and sets up a stage for further studies.

There is no doubt that this article should be published. It is original, of scientific significance, comprehensive, and well written. I have only some minor remarks and would like to ask the authors to address them while editing the final version of the manuscript.

Requested changes

Although the list is quite long, all of the points are minor. It is often up to authors' judgement whether the proposed changes are worthwhile to implement

Citations:

1- References [151],[158] do $gl(3)$ case only 2- [157] has any compact representation, not only rectangular ones. 3- Given that your citations are comprehensive, citing also 0706.3327 for Analytic Bethe Ansatz is probably worthwhile. 4- For Q-operators in supersymmetric case, also 1012.6021 in addition to what you have currently 5- In your review of supersymmetric spin chains (page 5/6), functional QQ-relations got explicit practical applications for these chains: 1608.06504, 1701.03704. 6- In relation to AdS/CFT: Review paper [136] is outdated, especially in relation to the spectral problem. Let me stress that the Hubbard model is applicable in the large volume asymptotic regime only (i.e. the sentence citing [136, 143] is a bit strong). The full spectral problem is based on $psu(2,2|4)$ symmetry and was solved by means of the quantum spectral curve: 1305.1939, 1405.4857. In reviews 1708.03648, 1911.13065 you can find more citations, while detailed relation of QQ-systems and QSC which is likely to be important for SoV is discussed in 1510.02100 7- If you are aware of any work where an analogue of inner boundary condition is written, please cite it. Currently, you cite [188,189,232] but it is not clear what exactly should the works be credited for. 8- Together with [189], this one - https://doi.org/10.1023/A:1025048821756

Statements/formulae in the article:

1- Twisted Yang-Baxter algebra can have different meanings (like twisted Yangian). So term is used a bit loosely 2- (2.9) is not a property of $gl(m|n)$ Lie superalgebra but of its fundamental (vector) representation. 3- Since dagger is often associated with Hermitian conjugation, it would be worth specifying in a footnote what l.h.s. of (2.18) means precisely. 4- Is it necessary (for immediately what follows after) that matrix (2.34) is restricted to the block-diagonal form? 5- Second paragraph of section 2.2, you presumably mean substraction of sets in the definition of a bidimensional lattice, then backslash sign. 6- “This orientation is consistent with the one used in [188]” -> This is false. Direction a is vertical in [188] and b (called s, for symmetrisation) is horizontal. 7- Strictly speaking, (2.49) is not satisfied in the (0,0) corner of the fat hook, provided convention (2.51) 8- "Let K be a (M+N )×(M+N ) square matrix solution of the gl(M|N) -graded Yang-Baxter equation of the block form (2.34),..." - from the context, it seems that K is a constant twist matrix. Why do you then say that it solves Yang-Baxter equation? It of course does, but in a very trivial way 9- In (2.90), l.h.s. is not defined 10- The authors might consider making title of section 3.2.2 more precise. Completeness is established for a specially degenerate twist, (and for generic enough inhomogeneities). Only after I read other parts of the paper could I understand that. For a reader who is skimming through, impression is that a much stronger statement is proven. 11- p.22 “in this case the transfer matrices T (K) are invertible” - please refer to a place in the text where it is demonstrated 12- In Conjecture 3.1, you say “Taken the general twisted…”. What these words mean precisely? Should we understand “the general” as “a” or “any” (hence you expect the conjecture to hold always), or “a generic” (hence you expect the conjecture to hold almost always, except for some bad values of twist, and there is no means to control which are these values, we only know that they are rare) 13- Based on the text, I understand that you give, on p.22, an argument in favour of the conjecture 3.1. Can you state precisely why is this argument not a proof yet? It seems you show that every solution (2.108), (2.109) gives $t_1$ as an eigenvalue. The other direction is obvious (Lemma 2.2). So it seems that both directions of iff in conjecture 3.1 are proven, but you don’t disclose all the details on p.22. 14- In the last paragraph of the proof to Theorem 3.1, you state that transfer matrix is diagonlizable which follows from Theorem 2.1. I presume you meant Proposition 2.1. 15- If I understood correctly, Corollary 3.1 relies on Theorem 3.1. Since Theorem 3.1 was stated a while ago in the text, could you please add “for almost any values of the inhomogeneities” in the text of the corollary?

Potential Grammar/typos noticed (it can be also a question of personal taste of course):

1 - Construction "A is something. While B is something else" with point is regularly used (sounds somewhat strange) 2- If eight-vertex model (with hyphen) then two-dimensional space, one-parameter deformation etc 3- "...of the separate variables..." -> of the separated variables 4 - "...a natural framework where to prove..." 5 - "...where the holding different representations..." (I do not understand the sentence) 6 - "Object that have a well defined parity, either even or odd, are called homogeneous" (objects or is called) 7 - "All these transfer matrices commutes" (commute) 8 - In conjecture 3.1: “define above -> defined above” 9 - In theorem 3.1: "...then the eigenvalue spectrum..." -> to drop "then"

---

## Round 3 · Author Response

Dear Editor,

We found very encouraging the overall positive description of our manuscript and results presented in the three referee’s reports. We would like to thank the referees for their attentive reading and understanding of the manuscript. Their numerous remarks, suggestions and identification of typos have given us the opportunity to improve the presentation of our paper and have been mainly all implemented in the current version of the manuscript. This is in particular the case for suggestions on the AdS/CFT framework and associated references that we have given in our introduction for completeness, but that we explicitly admit does not belong to our main field of expertise. The answer to referee’s remarks and suggestions is given together with the list of changes done in the improved version of our paper.

Sincerely yours,

J. M. Maillet, G. Niccoli, L. Vignoli

---

## Round 3 · List of Changes

# Answer to Referee 2 remarks and suggestions:

1. We have added the two references cited in the point 1 of the report, plus a reference to a further paper of Mukin et al. Moreover, we have modified the last paragraph at page 6 and added the footnote 6 to make more explicit the difference between SoV and the Bethe Ansatz methods. The main point is that SoV is not an ansatz but a constructive resolution method; the form of the wave functions of the transfer matrix eigenvectors in the SoV basis is not an ansatz, rather, it is completely characterized by the SoV basis itself. So the main problem to solve in our approach is the construction of such SoV bases. This makes the proof of the completeness of the spectrum description very clear and even easy in general as this is mainly a built-in feature of the SoV approach. One has just to properly identify closure relations (e.g. the quantum determinant condition for the fundamental non-supersymmetric representations) that enable to act with the transfer matrix on the constructed SoV basis in a close form. Then all the transfer matrix eigenvalues are solutions of these relations and vice versa one can show that all the nonzero solutions to this system define transfer matrix eigenvalues and nonzero eigenvectors by computing the action of the transfer matrix on the constructed factorized vectors in the SoV basis. Note that for the supersymmetric case we haven’t yet computed explicitly these actions for general twists, and this is why it imposes us to keep our claim on the spectrum as a conjecture.
Instead if one uses ansatz methods, one has to first show that the constructed potential eigenvectors are indeed nonzero. In NABA this is a priori a non-trivial task. For example, to do so and to our understanding, one establishes the isomorphism between the “good” Bethe ansatz solutions and the solution of the QQ-functional equations. Then one has to show that the ansatz is complete; i.e. that all the transfer matrix eigenvectors have the form given by the Ansatz. This is generally done by counting the number of solutions, which must coincide with the dimension of the quantum space if the transfer matrix is diagonalizable. This counting is not required in SoV, because we know that the factorized form of the transfer matrix eigen-wavefunction applies to any transfer matrix eigenvector by the construction of the SoV basis (where the main point is to prove that it is indeed a basis of the Hilbert space).

2. We have taken into account the point 2 of the referee by adding the footnote 8.

3. We have implemented all the suggestions and remarks of the point 3 of the referee.

4. We have added the reference math-ph/1401.7691 to our comment on the su(2|2) symmetry at the beginning of page 5.

5. We have added the citation to paper hep-th/1501.04651.

6. We have explicitly stated “large inhomogeneities limit”.

7. To our knowledge, the quantum spectral curve is a notion and terminology introduced by Sklyanin, see for example hep-th/9212076. As explicitly written in the abstract of Sklyanin’s paper, “the canonical coordinates and the conjugated operators are constructed which satisfy the quantum characteristic equation (quantum counterpart of the spectral algebraic curve for the L operator)”. Hence, the characterization of the spectrum in SoV is given by the quantum counterpart of the spectral curve of the monodromy operator. Note that the canonical coordinate operators are the operator zeros of the Sklyanin’s B-operator, while the (exponential of) canonical conjugated operators are the shift operators on the spectrum of the canonical coordinates. In general, we write the quantum spectral curve in its coordinate form, i.e. our quantum spectral curve can be seen as the matrix element of the Sklyanin’s one between a transfer matrix eigenstate and an SoV basis element, when Sklyanin’s SoV applies, otherwise our results generalize Sklyanin’s ones. However, in general, the fact that we can prove that the transfer matrix has simple spectrum and is diagonalizable allows us to rewrite these quantum spectral curves at the operator level, involving the quantum spectral invariant operators (i.e. the transfer matrices) and just one Q-operator.

8. In fact, for the special non-invertible twist for which the quantum spectral curve (3.26) is written, we have that the $\phi_t(\lambda)$ are directly related to the eigenvalues of the $Q_2(\lambda)$ operator, as evidenced by the equations (3.54) and (3.58). However, there is in general a non-trivial normalization to consider in (3.54). Moreover, one should remark that while the quantum spectral curve (3.26) holds only for the special non-invertible twist the formula (3.29), it can be a priori extended to a general twist with nonzero $ k_1$ and with $\bar\alpha=k_1$. This can be argued rewriting the transfer matrix eigen-wavefunctions in terms of the eigenvalues of the $Q_1(\lambda)$ operator by the NABA expression (3.45). For these reasons we have decided to change the notation of the zeros in $\nu$ in (3.25).

9. We agree with the referee and we have added the footnote 20.

# Answer to Referee 3 remarks and suggestions:

Concerning the link to QQ-systems, please see above comments about the relation between SoV and other methods in our answer to referee 2 (points 1 and 8). In general, QQ-systems provides necessary conditions for the eigenvalues (in the case they are derived algebraically from the R-matrix structure of the model), but it is a non-trivial task to determine which sets of solutions to such QQ-systems indeed correspond to true eigenstates. The SoV method is designed to solve such a problem. We have described the link in particular cases (see point 8 in the answer to referee 2) in the section 3.2.2. See also Appendix A.

We have also implemented the citations and clarifications that the referee has described in the section “Citation” of his report.

1. We agree with the referee and changed for the expression « twisted boundary condition » in the new version.

2. Indeed, we did not make a good distinction between the abstract algebra and its fundamental representation over C^(M|N) when introducing it. We have separated the two equations and we hope it's now clearer.

3. We have transposed most of the usual notations from the non-graded case to the graded one. For example, we are using « [ , ] » for the graded commutator, rather than the also common « [ , } ». For this reason, we keep \dagger for the Hermitian conjugation, which requires an adapted definition when being extended to a tensor space. We have added equations (2.18), (2.20) and (2.21) to clarify and illustrate the original definition (2.19).

4. The relation RKK = KKR is true for more general K matrices, but this requires introducing the supergroup GL(M|N) in details. However, since we had no use of non-diagonal twists in the present paper, we decided not to make the introductory content heavier by describing it there.

5. We agree with the referee and have modified the text accordingly.

6. We agree with the referee and have modified the text accordingly by clarifying the correspondence.

7. We agree with the referee and have modified the text accordingly.

8. The relation RKK = KKR encodes the GL(M|N) invariance of the R matrix and as such is non-trivial. It can also be interpreted as the Yang-Baxter relation for K with a trivial, one-dimensional, quantum space. However, it is a necessary requirement for the twist K such that the monodromy matrix with twisted boundary conditions given by K still satisfies the Yang-Baxter algebra.

9. We thank the referee for pointing out the poor introduction of the notation \bra{S,a}. In fact, we have reworded the statement of the theorem 3.1, as well as part of its proof relying now on the equation (2.91). We hope it’s now clearer how this object appears and how it is involved in the calculations.

10. We have thought that, being subsections of the section 3.2, the fact that the statement refers to non-invertible special case was evident. However, we agree with the referee that it is better to clarify this point. We have modified accordingly the titles of sections 3.2.1 and 3.2.2.

11. The transfer matrix is invertible in the inhomogeneities as by the reconstruction of the local operators [87,88], it is known that in the inhomogeneity $\xi_n$ this transfer matrix just coincides with the local matrix K at the site n dressed by the product of shift operators along the chain. So, it is invertible as all these operators are invertible. We have added a footnote to explain it.

12. We expect that the claim in the Conjecture 3.1 holds always for a nonzero K matrix. In the future we should be able to prove it for any twist for which it is possible to construct an SoV basis, i.e. for K being a simple matrix. Such proof by continuity arguments should lead to the proof that Conjecture 3.1 holds for almost any choice of the matrix K, i.e. up to a sub-variety in the space of the parameters of the these gl(M|N) matrices. So, there still remain some steps to prove it. We have added a footnote on this.

13. Indeed, there are some technical details that have to be developed to prove Conjecture 3.1 and, as mentioned above, we intend to do it in a further paper. The main missing point is the detailed computation of the transfer matrix action on a state who’s factorized wavefunctions in the SoV basis are written in terms of solution to the system of equations (2.108) and (2.109). The difficulty here is that the closure relations are of a higher degree compared to the standard quantum determinant relations we used in the gl(n) case. This closure relation is required to prove that these states are always transfer matrix eigenstates. Currently we have given only an argument on how this should be the case.

14. We agree with the referee and we have modified the text accordingly.

15. We agree with the referee and we have modified the text accordingly.

You are currently on this page

Resubmission 1907.08124v3 on 12 August 2020

---

## Editorial Decision

published